# Anthropogenic influence on extreme precipitation over global land areas seen in multiple observational datasets

Gavin D. Madakumbura [1✉], Chad W. Thackeray [1], Jesse Norris [1], Naomi Goldenson [1] & Alex Hall [1]

The intensification of extreme precipitation under anthropogenic forcing is robustly projected by global climate models, but highly challenging to detect in the observational record. Large internal variability distorts this anthropogenic signal. Models produce diverse magnitudes of precipitation response to anthropogenic forcing, largely due to differing schemes for parameterizing subgrid-scale processes. Meanwhile, multiple global observational datasets of daily precipitation exist, developed using varying techniques and inhomogeneously sampled data in space and time. Previous attempts to detect human influence on extreme precipitation have not incorporated model uncertainty, and have been limited to specific regions and observational datasets. Using machine learning methods that can account for these uncertainties and capable of identifying the time evolution of the spatial patterns, we find a physically interpretable anthropogenic signal that is detectable in all global observational datasets. Machine learning efficiently generates multiple lines of evidence supporting detection of an anthropogenic signal in global extreme precipitation.

[1] Department of Atmospheric and Oceanic Sciences, University of California—Los Angeles, Los Angeles, CA, USA. ✉email: gavindayanga@ucla.edu

Extreme precipitation can have devastating direct societal impacts such as flooding, soil erosion, and agricultural damage[1], as well as causing indirect health risks and impacts[2]. Anthropogenic warming acts to intensify Earth's hydrologic cycle[3]. This intensification is manifested in part through increased extreme precipitation as a result of greater atmospheric moisture with warming following the Clausius–Clapeyron relationship. However, circulation changes can act to enhance or reduce this increase[4–7]. Future projections by climate models following climate change scenarios show a robust increase in extreme precipitation, globally and on regional scales[8–11]. Moreover, increased variation between wet and dry extremes is projected, which could have devastating societal impacts[12,13]. These changes in extreme precipitation may have already become apparent on a regional basis[14–16].

Recent studies have detected anthropogenic influence in historical changes to extreme precipitation across the domains of North America[17,18], Europe[18,19], Asia[18–20], and Northern Hemisphere land areas as a whole[21]. These attempts are part of a larger category of studies known as Detection and Attribution (D&A)[22–24]. Often, they initially extract the spatial or spatio-temporal patterns of climate-system response to anthropogenic forcing (so-called fingerprints) from an ensemble of global climate models (GCMs). Projection of observations onto these fingerprints allows for detection of the signal[24,25]. The presence of a signal that can be statistically distinguished from internal variability confirms the influence of external forcing. Thus, traditional D&A methods rely on long-term observations[24,26]. In the case of extreme precipitation, traditional methods may be difficult to apply globally due to inordinately short records and large observational uncertainty, reflected in multiple global datasets produced with very different assumptions[27–30]. Another key difficulty with traditional methods is that the models produce a large spread in the extreme precipitation response to historical anthropogenic forcing[31]. This spread, the model uncertainty, occurs alongside large internal variability in the models' simulations of the historical period. These two effects create significant uncertainty in the character of the true anthropogenic signal. In past research, spread in the response has been suppressed by assuming the anthropogenic fingerprint can be derived from the ensemble-mean change in extreme precipitation[32]. Here, we aim to take these uncertainties fully into account, by making no assumptions about how to derive the anthropogenic signal from GCM data.

Machine learning-based methods for the detection of anthropogenic influence (DAI) have been shown to overcome the reliance on trends[33,34] and are even capable of detecting the human influence from weather data on a single day[35]. An artificial neural network (ANN) is trained to predict a proxy of external forcing (e.g., the year of the data) based on the spatial maps of the target variable from an ensemble of GCM simulations. Under this supervised learning approach, the ANN learns the spatial patterns that best represent the external forcing from the background noise arising from the internal variability and model uncertainty[33,34]. Observations can then be fed to this trained ANN to assess the presence of an anthropogenic signal in observations[33–35]. This ANN DAI method can identify the nonlinear combinations of the forced signal, internal climate variability, and intermodel variability[34]. This method also has the advantage of being able to explicitly include internal variability and model uncertainty. It does not assume that any model or any model-derived quantity, such as the ensemble mean of the models, is the true anthropogenic signal. It uses the raw GCM data, with GCM internal variability included. In addition, novel visualization techniques allow for the interpretability of the ANNs formerly considered as black boxes, making them explainable[36,37], or interpretable in terms of physical processes or system behavior. The use of these visualization techniques alongside the ANN DAI method allows one to capture the time-varying dynamic fingerprints of each input and evaluate their physical credibility[34,38].

In this study, we apply the ANN DAI method and the ANN visualization technique known as Layer-wise Relevance Propagation (LRP)[39,40] to global maps of annual maximum daily precipitation (Rx1day) over land. Using Coupled Model Intercomparison Project, phase 5 (CMIP5)[41] and phase 6 (CMIP6)[42] model ensembles, we first aim to understand how the ANN is detecting the anthropogenic signal and interpret it physically. Then we use the ANN to detect the anthropogenic influence on Rx1day in several land-only observational and reanalysis datasets. Thus, we are agnostic about which GCM is correct, and which gridded dataset is a true representation of the observed record. In this way, we efficiently generate multiple lines of evidence as to the presence of an anthropogenic signal in the various instantiations of the observed record.

## Results

**ANN-identified fingerprints of anthropogenic influence.** We first discuss the ability of the ANN to predict the year of occurrence for a series of simulated annual Rx1day maps. Predictions of the simulated Rx1day year (Fig. 1a, b) show that the ANN struggles during roughly the 1920–1970 period. But prediction accuracy gradually increases, noticeably starting from the late twentieth century. This characteristic, a near-constant predicted year followed by a positive trend, is consistent with the emergence of the anthropogenic signal from the noise of natural variability[43]. Compared to when this technique is applied to global-mean temperature (ref. [33]), there is a lag in the emergence of the anthropogenic signal in extreme precipitation. This delay is likely due to larger internal and intermodel variability in extreme precipitation. We estimate this time of emergence (departure year) as the year after which the ANN prediction continuously exceeds a selected base period (1920–1949)[33,43]. In GCMs, the predicted year departs from the base period in the 1970s, but the departures mostly occur later, with lower and upper quartiles of 1993 and 2014, respectively (Fig. 1c). The ANN suggests that there is a detectable anthropogenic signal in the GCM's Rx1day during the historical period, consistent with traditional statistical methods[44].

Figure 1d shows the importance of each grid box for the ANN to identify the anthropogenic signal (hereafter called relevance patterns, see "Methods"), averaged over the period 1982–2015. Positive (negative) values in the relevance patterns correspond to an increase (decrease) in the predicted year. Therefore, areas of positive relevance can be interpreted as the regions with a positive contribution to the prediction (i.e., the year) and negative values are the regions with a negative contribution. The sum of all grid cell values is equal to the predicted year ("Methods" and Supplementary Text). By learning how to predict the year of the data, the ANN is able to detect the spatial patterns that best reflect the changing climate from background noise[33,34]. Therefore, the relevance patterns observed above can be considered as the ANN-identified fingerprints of anthropogenic influence on Rx1day (e.g., ref. [35]).

The regions with positive relevance include the East Asian and African monsoon regions, and the North Pacific and Atlantic storm tracks (Fig. 1d and Supplementary Fig. 1). The regions with negative relevance include arid and semi-arid subtropical zones such as Northern African and Middle Eastern deserts, Southern South Africa, Australian arid and semi-arid regions, and wet regions such as central and northwestern parts of South America.

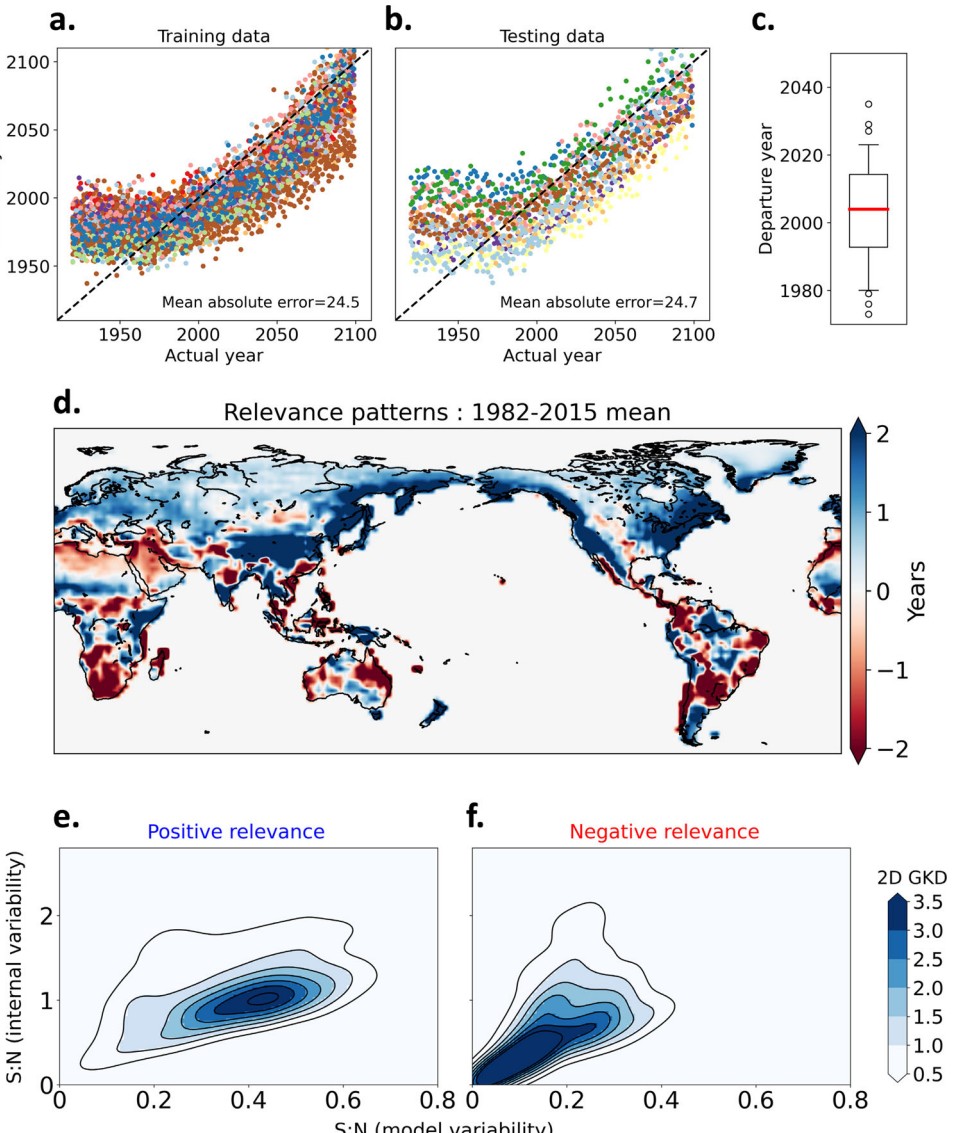

**Fig. 1 Fingerprint of external forcing in simulated Rx1day learned by the neural network. a**, **b** Actual year vs predicted year for training data derived from CMIP5 and CMIP6 global climate models (GCMs) (**a**) and testing data derived from CMIP5 and CMIP6 GCMs (**b**) for a single neural network. Each GCM is represented by a different color. **c** The year of departure from the base period, 1920–1949. Whiskers represent the 5th–95th percentiles, while blank circles represent outliers. **d** Multimodel, ensemble-mean, layer-wise-relevance-propagation-based relevance maps for annual maximum daily precipitation (Rx1day) input for the period 1982–2015 from all models. Relevance is the contribution of each grid value to the neural network's decision (see "Methods"). **e**, **f** Signal-to-noise ratio (S:N): two-dimensional Gaussian kernel density (2D GKD) estimation plots for grid cells with a positive relevance (**e**) and negative relevance (**f**) in panel (**d**). Signal is defined as the multimodel mean change in Rx1day between the base period 1920–1949 and 2070–2099. Noise is defined in two ways: the first stems from internal variability and is calculated as the multimodel ensemble mean of the standard deviation in Rx1day during the base period. The second pertains to intermodel variability and is calculated as the intermodel standard deviation of the signal from each GCM.

Regions with negative relevance coincide with areas where the dynamical component of the Rx1day trend (i.e., the contribution from the change in vertical velocity[4]) is largely negative (ref. [45], their Fig. 3b). This offsets the Rx1day increase stemming from the thermodynamic contribution (i.e., the contribution from the increase in atmospheric moisture with warming[3–5]) and produces only a weak and inconsistent increase in Rx1day[45]. The uncertainty associated with the dynamical component has been identified as a major concern for D&A of precipitation[46].

To understand the physical nature of the relevance patterns, we next assess the signal, and the noise components arising from internal variability and the model uncertainty. The negative relevance of the forced response is associated with a lower signal-to-noise ratio (S:N) than the regions with positive relevance

(Fig. 1e, f). The S:N is lower for both internal variability and model variability. This reflects both the higher uncertainty regarding the change in extreme precipitation projected by GCMs for a majority of global arid land regions, as well as larger internal variability in those regions.

The ANN-based relevance patterns are consistent with the idea that previously observed long-term trends of terrestrial Rx1day are anthropogenic in origin (e.g., ref. [21], their Fig. 1). Many wetland regions, such as the Asian, African, and South American monsoon regions, have experienced a robust increase in Rx1day to date[15,16], whereas in arid and semi-arid subtropical zones no such trend can be seen[16]. The selection of regions in these previous studies (e.g., ref. [16]) seems to overlap with regions of higher relevance in Fig. 1d.

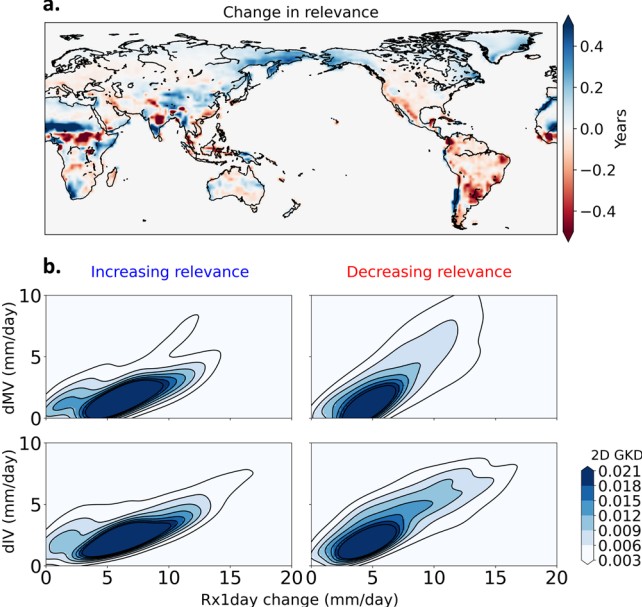

**Fig. 2 Change in the relevance patterns learned by the neural network through time. a** Multimodel average change of relevance maps between 2070–2099 and 1920–1949. Relevance is the contribution of each grid value to the neural network's decision, obtained using layer-wise relevance propagation (see "Methods"). **b** Multimodel ensemble-mean change in annual maximum daily precipitation (Rx1day) vs change in intermodel variability of Rx1day, (dMV, top panels), change in Rx1day vs change in internal variability of Rx1day (dIV, bottom panels), between 2070–2099 and 1920–1949. Two-dimensional Gaussian kernel density (2D GKD) estimation is shown. Left panels show results for grid cells where relevance increases with time in panel (**a**) and right panels show results for grid cells where relevance decreases. Internal variability is calculated as the standard deviation of Rx1day time-series and intermodel variability is calculated as the standard deviation of mean Rx1day from all models for each time period. Prior to the calculation of internal variability, the forced Rx1day trend at each grid cell was removed by regressing onto 41-year lowess filtered annual global-mean surface temperature[85].

**Time-varying fingerprints**. One of the main advantages of using an ANN to detect anthropogenic influence over traditional D&A methods is that time-varying signals can be accounted for[34,38]. Changes in the signals could be due to the nonlinear evolution of the climate system or temporal and spatial variations in the forcing itself. Figure 2a shows the difference between the relevance maps for our baseline period (1920–1949) and the end of the twenty-first century (2070–2099). While the sum of the relevance maps derived using LRP is larger for later years in the time series ("Methods"), local differences can explain the redistribution of the importance with time. This ability to aggregate over regions and different samples has been identified as an advantage of using LRP to interpret deep-learning models[40]. Notably, the relevance increases with time across Africa and Asia, which is likely to be associated in part with the enhancement of the monsoon systems[47]. A similar increase in relevance can be seen in North Pacific and North Atlantic land regions, possibly associated with the poleward shift of storm tracks[48]. South African and South American Mediterranean climate regions also show an increase in relevance, associated with subtropical drying, a robust pattern of climate change[49,50]. This indicates that even though dry regions have a smaller S:N compared to wet regions in terrestrial Rx1day (Fig. 1d–f), some dry regions show an increase of the signal and/or decrease in noise with time, yielding an increase in the relevance (Fig. 2a). Among the regions with

decreasing relevance, a majority of South America and the Western US stand out, possibly due to an increase in model uncertainty of Rx1day as the twenty-first century progresses.

To assess the physical validity of the change in relevance determined by the ANN, we break the terrestrial Rx1day record down into its forced signal and changes in noise components between the two periods. Results show that grid cells with increasing relevance have a comparable change in Rx1day, but much less increase in both internal variability and intermodel variability, compared to grid cells with decreasing relevance (Fig. 2b). Therefore, the change of relevance over time is in accord with the tradeoff between increasing noise and increasing signal with time.

**Origins of the spread in the predicted year**. We next investigate why the ANN predicts such a large range of years depending on the data of the underlying GCM used to predict the year. This intermodel spread in the predicted year is especially pronounced before the warming signal emerges (Fig. 1a, b). Here, we select four GCMs with the highest average predicted year, and four GCMs with the lowest average predicted year, during the baseline period (1920–1949). We obtain the relevance heatmaps for each year of the baseline period for these eight models and calculate the composite difference (i.e., models with high-versus-low predicted year; Fig. 3a). Large positive values are seen in the African and Asian monsoon regions. The models predicting later years also have larger twentieth-century mean state Rx1day values in these regions (Fig. 3b). Thus, the GCMs that predict a later year in the baseline period have more future-like patterns of Rx1day in their baseline climatologies compared to other models. When projected onto the fingerprints identified by the ANN, these patterns result in a later predicted year compared to the opposite subset. This exercise suggests a potential use of ANN-based DAI methods to understand how biases in historical simulations project onto future changes[31].

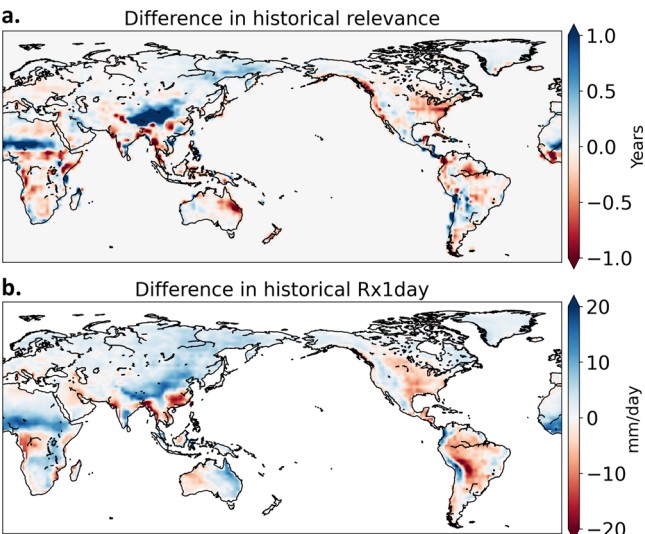

**Fig. 3 Differences between subsets of models with high and low predicted years by the neural network during the baseline period (1920–1949). a, b** The difference in their relevance maps (**a**) and annual maximum daily precipitation (Rx1day) (**b**) between the four models with the highest mean predicted year and the four models with the lowest mean predicted year (as shown in Fig. 1a, b). Relevance is the contribution of each grid value to the neural network's decision, obtained using layer-wise relevance propagation (see "Methods").

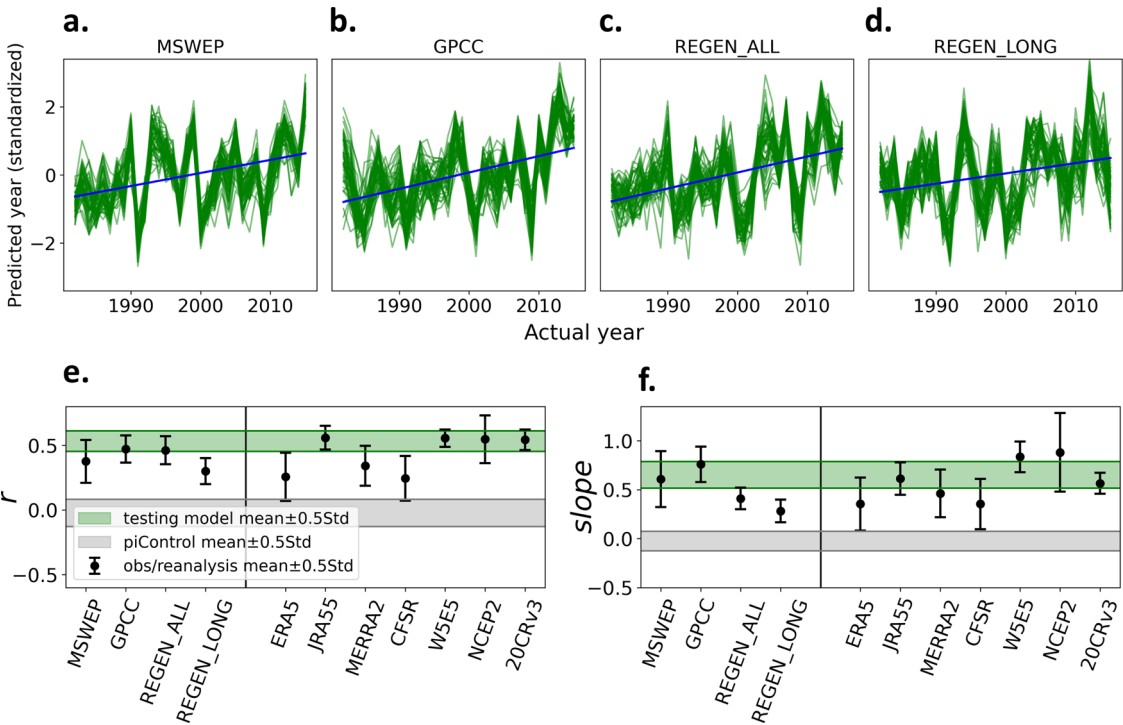

**Fig. 4 Metrics of the forced signal in observation-based estimates of precipitation during 1982–2015. a–d** Actual year vs predicted year obtained from 51 different artificial neural networks (ANNs) with different training/validation/testing sets, for four observational datasets, MSWEP (**a**), GPCC (**b**), REGEN_ALL (**c**), and REGEN_LONG (**d**). Green lines show results from each ANN. The blue line is the mean slope. Each predicted year time series is standardized in the figure for a better comparison between datasets. **e** Correlation ($r$) between the actual years and predicted years, (**f**) slope of the regression line between actual years and predicted years for observational and reanalysis data (black circle with a line), and testing models (green-shaded regions). Gray shading represents a measure of natural variability derived from 220 nonoverlapping 34-year segments obtained from pre-industrial control (piControl) simulations (see "Methods"). The error bar range and the range of the green and gray-shaded area show the ±0.5 standard deviation range.

**Detected anthropogenic signal in historical Rx1day records.** With these physical interpretations of the ANN results and relevance patterns, we use the GCM-trained ANNs to detect whether there is a forced signal in observations. According to previous studies, a steady global warming trend can be seen since the 1970s[51] and, in GCMs, the anthropogenic signal of global-mean Rx1day has started to emerge as of the 1970s[44], Therefore, according to the theoretical basis of the response of extreme precipitation to warming[3–5], one could hypothesize that GCM-simulated and observed Rx1day should have a positive significant trend during the historical period analyzed here, 1982–2015. Confirming this, GCMs show a positive trend in globally averaged Rx1day (significant at 99% in 36 out of 44 models), which cannot be explained by natural variability alone (Supplementary Fig. 2). In observations and reanalyses, only seven out of the eleven datasets show a significant trend (significant at 99%) in globally averaged Rx1day for the historical period 1982–2015, ranging from 0.02 to 0.09 mm/day/year (Supplementary Table 1). Taken at face value, this large disparity in observations suggests that the observational evidence for anthropogenic influence on recent changes in extreme precipitation is weak. However, when we apply the ANN trained on Rx1day data from GCMs, to the same eleven datasets, a different story emerges.

If an observational dataset exhibits the same forced response as the GCMs, the time series of predicted year from that dataset should have a positive correlation ($r$) with the actual year and linear regression of these two variables should produce a positive slope[25,34]. The metric $r$ can be considered as an indicator of the presence of an anthropogenic signal whereas the slope is an indicator of the strength of that signal. Figure 4 shows these two metrics for observations, reanalyses and testing GCMs, from 51

different ANNs trained using randomly selected training GCMs. We also calculated the two metrics ($r$ and slope) for the predicted-versus-actual year given for GCM simulations with radiative forcing held constant at pre-industrial levels, which is used as a measure of natural variability (see "Methods"). All observations and reanalysis have positive $r$ values (Fig. 4 and Supplementary Fig. 3), even in datasets that do not show a significant positive trend in global-average Rx1day (Supplementary Table 1). This contrast is because the ANN detects a signal in the spatial distribution of Rx1day, as opposed to the global average. The $r$ values for all observational datasets are substantially larger than those expected by natural variability (gray-shaded area in Fig. 4e, f). When looking at the slope, two observational datasets (MSWEP and GPCC) are in line with GCMs, along with four reanalyses (JRA55, MERRA2, W5E5, and 20CRv3). The two REGEN datasets, ERA5 and CFSR, show lower slopes, whereas NCEP2 has the highest slope among the datasets considered here. In general, observational and reanalysis products show similar $r$ values and slopes as the GCMs for the same historical time period (compare the black circles and the green bands in Fig. 4e, f). This indicates that the observational and reanalysis products show anthropogenic influence on Rx1day that is comparable to what is shown by GCMs.

To estimate the statistical significance of the signal detected in the observations and reanalysis, we first estimate the noise as the standard deviation of the distribution of the slopes representing natural variability (Fig. 4e, f). Then S:N is calculated in all datasets by dividing the mean slope by noise. Following the two-tailed z test, S:N larger than, for instance, 1.96 corresponds to a statistical significance level of 95%[25,52,53]. Out of the four observations, MSWEP, GPCC, and REGEN_ALL show a 95% significance and

REGEN_LONG shows a 84% significance. Among reanalyses, ERA5 and CFSR show a 90% significance while the rest show a 95% significance.

These results demonstrate that the absence of a significant linear trend in globally averaged Rx1day cannot be taken to mean there is no evidence of an anthropogenic signal in Rx1day. This underscores the importance of exploiting the spatial pattern of the response to external forcing to extract the forced signal in observations, as opposed to the trend-based analysis[35,38,54]. In particular, areas with higher internal variability can act to suppress the trend in the global mean. Further evidence of the importance of spatial patterns can be seen in the fact that the average ANN-predicted values vary widely and systematically across the observational datasets (Supplementary Fig. 3). This is an indicator of systematic and large relative biases in the Rx1day climatologies of the various datasets (as pointed out above in the discussion of ANN applied to the GCMs, the average predicted value of the year depends on the magnitude of the Rx1day in the climatology (Fig. 3 and Supplementary Text)). Yet it is significant that the ANN can put the years in close to the correct order, as demonstrated by the significant correlations between actual and predicted years, even if the absolute value of the years is incorrect. This is a strong indicator that the subtle patterns and time variations of the simulated anthropogenic signal are present in the observational datasets and are shared among them, despite the fact that they are systematically biased relative to one another and likely the real world[55,56].

## Discussion

Detecting anthropogenic signals in observations of extreme precipitation has been a challenging task due to the large internal variability of rare events, as well as climate model uncertainty. The limited sampling in observations adds additional uncertainty, due in part to a dataset development process that involves a variety of homogenization, extrapolation, and interpolation techniques to produce global gridded products[30]. Using a recently introduced ANN DAI method, we utilized the time evolution of spatial maps of Rx1day in GCMs, for historical simulations and future projections. The ANN yields fingerprints of anthropogenic signals that are physically consistent with the time evolution of the forced signal and can be distinguished from the noise arising from internal variability and substantial model uncertainty. Using this ANN DAI method, we show that the anthropogenic signal can be detected in all global terrestrial Rx1day records considered in this study. This robust detection occurs despite large systematic biases and large discrepancies in data sources and homogenization methods.

While previous trend-based D&A assessments of Rx1day have demonstrated the human influence in this variable in some regions, those studies assume the ensemble mean of the GCMs is the anthropogenic signal. This leads to questions as to whether further steps are needed to fully consider model uncertainty[32,57]. We made a simple attempt to examine this issue by applying the ANN DAI method to the same widely used, quality-controlled Rx1day record used in the previous trend-based D&A assessments. We applied the method twice, once using the same multimodel approach discussed elsewhere in this study, and once using a large ensemble dataset which only accounts for internal variability. Our results show that including internal variability and model uncertainty in the forced response could reduce the power of detection (Supplementary Text). Therefore, the detected signal in multiple global terrestrial Rx1day datasets in this study, with internal variability and model uncertainty being taken fully into account (Fig. 4), is a definitive affirmation of a human influence on extreme precipitation in the historical record. Note that while all observations show this anthropogenic influence, the signal magnitude varies considerably, on par with that seen in the GCMs. This large observational uncertainty underscores a difficulty in constraining future projections of extreme precipitation with historical climate model simulations and observations[31,58].

Several caveats of the machine learning-based detection method should be noted. Compared to regression-based traditional D&A methods[59], the assessment of the influence of individual forcings (e.g., anthropogenic aerosols, land-use change, and natural forcings such as volcanic and solar activities) in the presented framework is challenging. We did not attempt such a breakdown in this study, and this would require methodological modifications[60]. In addition, the training GCMs might be undersampling the low-frequency natural variability such as Atlantic Multidecadal variability and Pacific Decadal Oscillation. This may be remedied by inflating the training dataset with paleoclimate data[61]. However, even with adequate sampling of natural variability in the training dataset, the underestimation of the precipitation response to natural forcings such as volcanic activities and natural variability such as El Nino Southern Oscillation in GCMs could still affect the results[62]. We also note that different ANN visualization techniques are available[63–65], and those should be explored to understand the sensitivity of the extracted fingerprints to the ANN visualization technique. Despite these limitations, it is clear that ANN DAI methods with ANN visualization techniques are very useful and efficient in identifying the human influence on variables that are highly uncertain in GCMs, and poorly characterized in observations, such as extreme precipitation.

## Methods

**Data.** We use daily precipitation rate output from a collection of climate models participating in CMIP5 and CMIP6 (Supplementary Table 2). Data from each ensemble's historical forcing scenario are combined with future projections following a high-emissions scenario to create a time series from 1920 to 2099 for each model. Future projections from CMIP5 follow the Representative Concentration Pathway 8.5 (RCP 8.5)[66], while CMIP6 projections follow the Shared Socio-economic Pathway 5–8.5 (SSP 5–8.5)[67]. To increase our sample size, we combine both CMIP5 and CMIP6 model subsets into one ensemble, which is justifiable considering the very similar time evolution of the total anthropogenic forcing in RCP 8.5 and SSP 5–8.5 scenarios (ref. [67], their Fig. 3c). We regrid all daily precipitation data to a 2° × 2° spatial grid and compute the Rx1day value for each year at each land-grid point.

To assess the influence of natural variability, we also use pre-industrial control simulations (piControl), which are GCM simulations with radiative forcing held constant at pre-industrial levels. As the length of the piControl simulations vary between GCMs, we selected the same number of samples from a collection of 20 CMIP6 models used here (Supplementary Table 3). We extract 34-year nonoverlapping samples from each simulation (so as to match the length of the observational record) to represent natural variability. Each GCM provided 14 samples of this length and after removing the first three samples of each simulation to avoid climate drift[42], we were left with 220 pre-industrial samples with which to assess natural variability.

We use four datasets of observational estimates of daily precipitation rate with global coverage: Multi-Source Weighted-Ensemble Precipitation, version 2 (MSWEP)[68], Global Precipitation Climatology Centre (GPCC) version 2018[69], and Rainfall Estimates on a Gridded Network (REGEN)[70], including both REGEN_ALL and REGEN_LONG. MSWEP is a hybrid reconstruction using in situ, satellite, and reanalysis data, whereas GPCC and the REGEN datasets are developed from ground-based measurements. REGEN_ALL is developed by interpolating all considered station data, whereas REGEN_LONG is developed using only the stations with a data record of 40 years or longer. We further use seven widely used reanalysis products for comparison: European Centre for Medium-Range Weather Forecasts ERA5[71], Japanese 55-year Reanalysis (JRA55)[72], Modern-Era Retrospective analysis for Research and Applications, Version 2 (MERRA2)[73], NCEP Climate Forecast System Reanalysis (CFSR)[74], the bias-corrected ERA5 precipitation dataset compiled for phase 3b of the inter-sectoral impact model intercomparison project (W5E5)[75,76], NCEP-DOE Reanalysis 2 (NCEP2)[77] and NOAA-CIRES-DOE Twentieth Century Reanalysis version 3 (20CRv3)[78]. These observational and reanalysis datasets are selected considering the availability of full global land coverage and data for at least three decades. We selected the period 1982–2015 for observational analysis as it is the common temporal range for all datasets. All observation and reanalysis data were

regridded to the same 2° × 2° spatial grid as the models, and then Rx1day was calculated at each grid point for each year.

**Neural network-based detection method.** Here, we apply the method in ref. [33] to predict the year with which given annual Rx1day maps from GCMs are associated, a regression task. This requires the ANN to learn the signature of the forced response in simulated Rx1day. By feeding the ANN data from forced simulations, it learns to distinguish the forced signal from internal climate variability. The use of multiple GCMs helps the ANN learn the common elements of the forced response most relevant to the prediction task, a process that fully considers model uncertainty as well as internal climate variability. Input to the ANN from each model is a vectorized spatial map of Rx1day (2° × 2° spatial grid = 16,200 grid values) for each year from 1920 to 2099. Our primary goal is to detect the anthropogenic signal in extreme precipitation over land (excluding Antarctica). Thus, we mask out data over the ocean at this stage, resulting 6082 land-grid values. The ANN architecture consists of two hidden layers with ten nodes each. The Rectified Linear Unit activation function is used for all hidden units.

Approximately 60% of the models (26) are used for training the ANN, while the rest of the models are divided equally to use as validation and testing sets (9 models each). The mean squared error between the actual and predicted year of Rx1day is used as the loss function to be minimized during the training. For the optimizer which updates the ANN based on the gradient of the loss, we select rmsprop. Climate variables inherently contain spatial autocorrelation. To account for this dependence among adjacent input data points, we use L2 regularization[79] between inputs and the first hidden layer, which adds the sum of squared weights as a penalty term to the loss function. By iterating over L2 values of leading order of magnitudes and inspecting the tradeoff between low-prediction error and generalizability (Supplementary Fig. 4), we found L2 = 0.001 to be a suitable value for our analysis. We trained the model for 1000 epochs. Early stopping was enabled to reduce the overfitting by monitoring the validation loss with a patience value of 50 epochs[80]. We repeated the training process for 51 different training sets obtained by random combinations of GCMs, resulting in 51 different ANNs. We found that increasing the number of hidden units or changing the other hyperparameters did not result in a substantial increase in accuracy.

**Neural network interpretation using layer-wise relevance propagation (LRP).** Assume that for a given input map, $\mathbf{x}$, we get an output $f(\mathbf{x})$, in our case, the predicted year. LRP conservatively back-propagates this value through hidden layers until it reaches the input map. This process generates a relevance heatmap, indicating the areas of importance influencing the value $f(\mathbf{x})$. The conservation property is shown in Eq.1, for relevance propagation between two hidden layers $j$ and $k$, where $k$ is the upper layer (i.e., closer to the output). $\sum_{i=1}^{d} P_i$ denotes the sum of the relevance of the $d$ input features. The summation operation for each hidden layer (e.g., $\sum_k P_k$) is the summation of the relevance ($P$) of all hidden units in that layer, where $P_k$ is the relevance of a single unit in layer $k$. The activation, $a_k$ (Eq. 2) is the information coming from all units in layer $j$, to a target unit in layer $k$. In Eq. 2, $a_j$ values are the individual activations of each unit in the layer $j$, $w_{jk}$ values are the weights associated with the relationship between each unit in layer $j$ and the target unit in layer $k$, and $b_k$ is the bias of that target unit.

$$\sum_{i=1}^{d} P_i = \ldots = \sum_j P_j = \sum_k P_k = \ldots = f(\mathbf{x}) \qquad (1)$$

$$a_k = ReLU\left(\sum_j a_j w_{jk} + b_k\right) \qquad (2)$$

$$P_j = \sum_k \left(\alpha \frac{a_j w_{jk}^+}{\sum_j a_j w_{jk}^+} - \beta \frac{a_j w_{jk}^-}{\sum_j a_j w_{jk}^-}\right) P_k \qquad (3)$$

The relevance-propagation rule from layer $k$ to a unit in layer $j$ is given in Eq. 3. This general form is also known as the $\alpha\beta$-rule[39,40]. The components $()^+$ and $()^-$ indicate only positive and negative parts are being considered, respectively. The $\alpha$ and $\beta$ coefficients represent the relative amount of positive and negative relevance to be propagated, respectively. As shown in Eq. 3, positive relevance (i.e., excitatory influence) and negative relevance (i.e., inhibitory influence) are associated with positive and negative weights, respectively. The $\alpha$ and $\beta$ coefficients are to be chosen with the constraints $\alpha - \beta = 1$ and $\beta \geq 0$. The combination $\alpha = 2$ and $\beta = 1$ (LRP$_{\alpha2\beta1}$) has been experimentally inferred as suitable, and has been adopted in previous research[39,40,81–83]. Here we adopt the LRP$_{\alpha2\beta1}$ rule (Supplementary Text).

## Data availability

CMIP data used are available at https://esgf-node.llnl.gov/projects/esgf-llnl/. Observational and reanalysis data used are available at the following links: MSWEP: http://www.gloh2o.org/, GPCC: https://www.dwd.de/EN/ourservices/gpcc/gpcc.html, REGEN_ALL: https://doi.org/10.25914/5ca4c380b0d44, REGEN_LONG: https://doi.org/

10.25914/5ca4c2c6527d2, ERA5: https://cds.climate.copernicus.eu/, JRA55: https://rda.ucar.edu/datasets/ds628.0/, MERRA2: https://disc.gsfc.nasa.gov/, CFSR: https://www.ncdc.noaa.gov/data-access/, W5E5: https://esg.pik-potsdam.de/, NCEP2 and 20CRv3: https://psl.noaa.gov/data/.

## Code availability

Neural network analysis was conducted using Python libraries TensorFlow (https://www.tensorflow.org) and Keras (https://keras.io). Neural network interpretation was carried out using the library iNNvestigate[84] (https://github.com/albermax/innvestigate). Python scripts developed for the analysis and figures are available publicly at https://doi.org/10.6084/m9.figshare.14479659.

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

## Acknowledgements

We acknowledge the World Climate Research Programme's Working Group on Coupled Modelling, which is responsible for CMIP, and we thank the climate modeling groups for producing and making available their model output. We also thank the Earth System Grid Federation (ESGF) for archiving the data and providing access, and various funding agencies who support CMIP and ESGF. We acknowledge support from the Regional and Global Model Analysis Program for the Office of Science of the U.S. Department of Energy through the Program for Climate Model Diagnosis and Intercomparison.

## Author contributions

G.D.M., C.W.T., J.N. and A.H. designed research. G.D.M. developed the detection framework, conducted the analyses, interpreted the results, and wrote the manuscript. C.W.T., J.N., N.G., and A.H. contributed to the interpretation of the results and edited the manuscript.

## Competing interests

The authors declare no competing interests.
