## [Peer Review File · Nature Communications]

Reviewer comments, first round -

Reviewer #1 (Remarks to the Author):

Reviewer Summary:

In this study, the authors use a new pattern recognition method from machine learning to identify forced signals in extreme precipitation. In particular, they use data from CMIP5/6 models and observations in order to identify an anthropogenic signal in annual daily maximum precipitation (Rx1day) over land (globally). The authors also use an explainable AI technique called layer-wise relevance propagation (LRP) to understand how their machine learning model is making its predictions. By comparing the LRP maps to standard signal-to-noise composites, they find regional patterns that are comparable to previous attribution methods and studies on biases in global climate models (GCMs). Using these tools, the authors claim that an anthropogenic signal in extreme precipitation (globally) is detectable in observations.

General comments:

Overall, this is a very interesting study due to the application of a novel method of detection and attribution. Given the high uncertainties in attributing extreme precipitation due to data uncertainties, this study will be of high interest to the climate community. It also fits well within the scope of Nature Communications. For the most part, the methods are clear and well-stated, and the figures are of high quality. I only have some minor suggestions and comments in regard to the setup of the artificial neural network (ANN) and the interpretation of physical mechanisms that may relate to the patterns on the LRP maps. There are also a number of typos throughout the text.

Recommendation:

Minor revisions

Specific points:

1. L10; Change "product" to "project"
2. L10-11; Do GCMs also simulate large increases in extreme precipitation for years that correspond to observations? Or are you describing future climate model projections?
3. L16-18; What do you mean here? By design, studies that use single-model large ensembles also assess internal variability to understand changes in precipitation.
4. L19; Observational datasets?
5. L22; "...anthropogenic [global] signal in extreme precipitation."
6. L30-32; Rewrite to improve clarity.
7. L36-37; Another reference may be helpful for more context here (e.g., Swain et al. 2020)
8. L41; Change "anthropogenic" to "external" forcing
9. L47-48; Model structural biases also contribute to the total uncertainty (emission scenario + model + internal variability)
10. L52-54; Change to "Machine learning-based methods for the detection of anthropogenic influence (DAI) have been shown..." – as there are slightly difference approaches in Barnes et al. (2019, 2020) and Sippel et al. (2020).
11. L56-57; Given that these are relatively new methods for the climate community, it would be helpful to further clarify how the forced signal is detected from the ANN setup.
12. L89-91; Be clear here on the use of the word "model" which can be used to describe the ANN (e.g., in computer science) or GCM (e.g., in climate science)
13. L92-94; I am not sure I understand why you picked the 1920-1949 period to composite for the LRP map here? Wouldn't it be more interesting to compare that base period to the LRP composite when the forced change is detected? I actually found the LRP result in Figure S4a more interesting than the one in the main manuscript.
14. L99-115; I think it would strengthen the manuscript to better contrast potential physical mechanisms that may relate to the LRP maps. In fact, if there is room in the word count, I suggest moving some of the text from the supplement into this section.
15. L104-105; Where is this shown?
16. L136; This section is an interesting result of these new methods that should be noted further

in the conclusions/abstract!

17. L145; What do you mean by "higher value?"
18. L162-163; State the years of the historical period.
19. L164-166; This transition makes it sound like you are inputting maps of extreme precipitation trends into the ANN.
20. L171; "... of the ANN with different combinations of training and model GCM data."
21. L174; "by [random] chance."
22. L175-176; What are in line? Their slopes?
23. 178; "reanalysis"
24. L185-186; But can we directly infer this from the negative relevance? Since LRP only tells us the areas of importance for the ANN to make a prediction, it could be the case that nonlinear combinations of different regions (positive + negative relevance) act to oppose/enhance trends.
25. L189-192; How does the skill of the ANN change if you remove the annual mean from each Rx1day map (e.g., as done in Barnes et al. 2019)? Thus, this could be a way to reduce the model mean state biases.
26. L215; What do you mean by realistic radiative forcing?
27. L229; Note that this is on a global mean scale.
28. L397-398; Do you think that this relative coarse grid could affect the results of your ANN design for extreme precipitation?
29. L418; Multicentury?
30. L423-424; Do you include polar regions, such as Antarctica?
31. L432-436; Reference would be helpful here for readers unfamiliar with L2 regularization.
32. L347; Reword this sentence, as it reads contradictory to your statement in L439.

Figures/Tables:

1. Figure 1; The color map needs to be changed in (a) and (b) to a colorblind friendly choice, instead of "jet/rainbow." For example, see Hawkins, 2015. The red dashed (1:1) line is also very difficult to see. There is a typo in the title under (a) for "training".
2. Figure 1; The caption for (d) seems different than the title. What LRP years are you compositing here?
3. Figure 2; Aren't the relevance map unitless? I am a bit confused by the title under Figure 2a for (years).
4. Figure 3; I found it difficult to compare the observations in Figures 3a-d due to the different y-axes limits.

References:

Hawkins, E. (2015). Scrap rainbow colour scales. *Nature*, 519(7543), 291-291.

Swain, D. L., Singh, D., Touma, D., & Diffenbaugh, N. S. (2020). Attributing extreme events to climate change: a new frontier in a warming world. *One Earth*, 2(6), 522-527.

Reviewer #2 (Remarks to the Author):

In this manuscript, Madakumbura et al. use a machine learning method to identify the anthropogenic influence on changes in extreme precipitation at the global scale. This is an important topic and this paper adds to the literature by using an entirely different method than most existing studies. I think it will be useful to those interested in either/both attribution and extreme precipitation. Given that what is new about this paper is in the details and the method, it could perhaps be better suited for a longer-form article. I would be interested to read more in-depth discussion of the results, including of the time-varying fingerprints. I have some comments below for the authors to consider.

The introduction mentions two studies detecting human influence on increases in extreme precipitation. However, there are some other papers published in the last year that should be acknowledged:

Paik et al. 2020: Determining the anthropogenic greenhouse gas contribution to the observed intensification of extreme precipitation. doi:10.1029/2019GL086875.

Dong et al. 2021: Attribution of extreme precipitation with updated observations and CMIP6 simulations. doi:10.1175/JCLI-D-19-1017.1.

Dong et al. 2020: Detection of human influence on precipitation extremes in Asia. doi:10.1175/JCLI-D-19-0371.1.

In addition to North America as already mentioned, anthropogenic influence has also been identified for Europe and Asia. I think the Paik et al. paper is cited in the supplementary material, but it should also be noted in the main text.

How would the results be impacted if Rx1day was locally-standardized first? This could effect the differences between the wet and dry regions. Also, for the global means calculated, averaging non-standardized values will give more weight to the wet regions.

Specific comments:

- Line 21: Can you please clarify or rephrase "trend of the projection"? Additionally, the optimal fingerprinting regression and the EOF-based fingerprinting approach as in ref 21 are different methods, with some different assumptions.
- Line 81: Can you state either the analysis period or which period the first 7-8 decades are here?
- Figure 1b: Why is the orange model so different from the others?
- Figure 1d and the corresponding discussion could use clarifying. From both the subplot title and Line 92, I was left wondering how you could consider this the anthropogenic fingerprint when it is based on such an early period. The caption mentions 2070-2099, but this should be noted in the text too.
- Line 97: It would helpful to say explicitly why using a time-based metric tells you about the anthropogenic forcing
- Line 164: Why do these trends imply the anthropogenic forcing is weak? Also, what about the historical trends over this shorter period tells us the anthropogenic influence?
- Line 169: Can you please clarify why this needs to be just a positive slope and not a slope consistent with 1?

Reviewer #3 (Remarks to the Author):

See attached.

[next page]

Review Madakumbura et al.

Key results

The authors use a relatively new machine learning technique, ANN, to find an anthropogenic signal in precipitation for multiple observation and reanalysis datasets, that accounts for internal variability and model uncertainty.

Validity

I am not aware of data/analysis related flaws that would prevent publication.

Significance

Results are significant.

Data and methodology

The method is applied to annual-maximum daily precipitation over land, and relies on changes in spatial patterns. My understanding is that ANN can better recognize the year of occurrence if there is anthropogenic forcing than if there is not because of the additional signal that can be learned.

I have limited background on both the methods and most of the numerous datasets used, as noted below.

Analytical approach

No comment. See below.

Suggested improvements

See below.

Clarity and context

There are several instances where I felt the clarity of the text could be improved to make this accessible to more readers. I have detailed these among my more specific comments provided below, but perhaps the most important is being more explicit about why ANN can identify the anthropogenic signal. I found that I required multiple reads, and comprehensive use of some of referenced articles to understand parts of this article.

References

Acceptable, to the best of my knowledge.

Your expertise

My primary expertise is in environments and dynamics of deep convection & extreme precipitation processes, specifically on meso- and convective scales. The specific details of the ANN method, GCMs, reanalysis, and global precipitation datasets are all outside of my expertise. Most of my commentary has focused on clarity/communication of findings.

I have provided a number of more specific comments below, most of which focus on clarity and context:

1. L10,L12 “Global climate models produce large increases in extreme precipitation when subject to anthropogenic forcing..” and “Models produce diverse precipitation responses to anthropogenic forcing” seem somewhat contradictory.
2. My takeaway was that changes in spatial patterns were really important to ANN’s ability to detect the changes, it would be nice if this could be worked into the abstract.
3. L56-59: Seems like three different versions of the same sentence (esp. the second and third)? Could they be combined?
4. L68: I believe the definition of Rx1day is annual-maximum daily precipitation?
5. Would it be possible to use known dataset traits (e.g. data availability/regional biases/etc) along with your results to gain additional insights?
6. L89: “According to the models...” seems vague, I assume you mean in this application of your analysis method to GCM projections. Is what you are trying to say that because there is a detectable signal in Rx1day in GCMs for present day using this method, it should therefore be able to detect a Rx1day signal of change in obs/reanalysis? Perhaps consider editing for clarity...
7. L92: While it’s defined in the Data/Methods section, I’d suggest defining relevance pattern here at the first use. This is a new term to me (and I suspect to most readers).
8. L94-96: Is there a simpler way to say ‘advancing tendency on the prediction (i.e. the year)’...?
9. L103-108: While connection to physical processes is important, it’s placement here seems abrupt and I find the transitions into and out of it difficult to follow. Though maybe it’s just the ‘also’ in line 103 that’s throwing me off at the start.

I felt like I needed to go read parts of the referenced paper to understand what the authors were trying to say here, so perhaps a bit more detail/explanation can be provided to alleviate that need. Specifically, this is the first mention of ‘dynamic’ and ‘thermodynamic’ components, which have very specific definitions in the referenced paper...

10. L108: “As suspected,” why was this suspected? I think the transition between the previous discussion and the signal-to-noise ratio could be made a bit more explicit.
11. L119-120: “The selection of regions in these previous studies (e.g. ref 16,17) seems to overlap’ – could you provide specific examples of apparent overlap so that readers don’t need to compare figures across multiple papers?
12. L145-146: Higher value of what?
13. What is ‘these physical interpretations’ referring to?
14. Did you notice any specific themes in peaks (Fig. 3a-d,) where ANN way over-predicts or under predicts the year (e.g. volcanic events like described in the Barnes et al study)?

15. Am I correct in understanding that it's actually the change in the ability of ANN to successfully predict the year over time that supports the finding of anthropogenic influence? In other words, this shallow version of ANN can not distinguish years based on spatial patterns of annual-maximum daily rainfall alone, until the impacts of anthropogenic forcing are seen, and the time that this occurs is considered the departure year...? It might be good to explicitly say this, maybe between the two sentences in line 82, since this is still quite a new technique (and it took me multiple reads & accessing Barnes et al. to sort this out). Is there a situation where it would be possible for ANN to successfully predict the years based on spatial patterns of Rx1day in the absence of anthropogenic forcing?
16. Fig 3 caption: What do you mean by randomly shuffling? Perhaps a better question, what is being shuffled?
17. Supplementary material 2nd to last paragraph: Could any known qualities about observational datasets explain something about these patterns (e.g. sparsity of obs)?
It might also be good to clarify here that you are no longer talking about the HadEX3 dataset, but the original observational datasets used...

Reviewer #1 (Remarks to the Author):

Reviewer Summary:

In this study, the authors use a new pattern recognition method from machine learning to identify forced signals in extreme precipitation. In particular, they use data from CMIP5/6 models and observations in order to identify an anthropogenic signal in annual daily maximum precipitation (Rx1day) over land (globally). The authors also use an explainable AI technique called layerwise relevance propagation (LRP) to understand how their machine learning model is making its predictions. By comparing the LRP maps to standard signal-to-noise composites, they find regional patterns that are comparable to previous attribution methods and studies on biases in global climate models (GCMs). Using these tools, the authors claim that an anthropogenic signal in extreme precipitation (globally) is detectable in observations.

General comments:

Overall, this is a very interesting study due to the application of a novel method of detection and attribution. Given the high uncertainties in attributing extreme precipitation due to data uncertainties, this study will be of high interest to the climate community. It also fits well within the scope of Nature Communications. For the most part, the methods are clear and well-stated, and the figures are of high quality. I only have some minor suggestions and comments in regard to the setup of the artificial neural network (ANN) and the interpretation of physical mechanisms that may relate to the patterns on the LRP maps. There are also a number of typos throughout the text.

We are extremely grateful to the reviewer for the thorough, encouraging, and constructive comments. We answered all the review comments and revised the manuscript accordingly. Please find the point by point answers below.

Recommendation: Minor revisions

Specific points:

1. L10; Change “product” to “project”
Done
2. L10-11; Do GCMs also simulate large increases in extreme precipitation for years that correspond to observations? Or are you describing future climate model projections?
Even for the historical period analyzed in this study (1982-2015), 82% of the CMIP models show a significant (at 99%) trend in Rx1day. We added this information to the Supplementary Table 2 and in a new figure (Figure S3).
3. L16-18; What do you mean here? By design, studies that use single-model large ensembles also assess internal variability to understand changes in precipitation.
Agreed. Removed “internal variability” from the sentence.

Original sentence: “Thus, previous attempts to detect human influence on extreme precipitation have not incorporated internal variability or model uncertainty, and have been limited to specific regions and observational datasets.”

Revised sentence: “Previous attempts to detect human influence on extreme precipitation have not incorporated model uncertainty, and have been limited to specific regions and observational datasets.”

[Revised lines 16-18]

4. L19; Observational datasets?

Yes. Changed “global datasets” to “global observational datasets”.

[Revised lines 20-21]

5. L22; “...anthropogenic [global] signal in extreme precipitation.”

Added “global” as suggested.

[Revised line 22]

6. L30-32; Rewrite to improve clarity.

Done. Revised as below.

Original: “If current warming trends continue, climate models project that the Earth’s atmosphere overall will move towards a more intense precipitation regime⁸⁻¹¹.”

Revised: “Future projections by climate models following climate change scenarios show a robust increase in extreme precipitation, globally and on regional scales⁸⁻¹¹.”

[Revised lines 31-32]

7. L36-37; Another reference may be helpful for more context here (e.g., Swain et al. 2020)

Done. Newly added the references *Paik et al., (2020)* and *Dong et al., (2020,2021)*. To provide more context, we also moved the IPCC chapter of detection and attribution (D&A), *Bindoff et al., (2013)* to the end of this sentence where we first introduce D&A (this reference was originally included in the next sentence).

[Revised lines 35-38]

8. L41; Change “anthropogenic” to “external” forcing

Done.

9. L47-48; Model structural biases also contribute to the total uncertainty (emission scenario + model + internal variability)

We modified the set of sentences as below to make it clear that we are talking about model uncertainty and internal variability in historical simulations. New additions are in red below.

“Another key difficulty with traditional methods is that the models produce a large spread in the extreme precipitation response to historical anthropogenic forcing³¹. This spread,

the model uncertainty, occurs alongside large internal variability in the models' simulations of the historical period. These two effects create significant uncertainty in the character of the "true" anthropogenic signal."

[Revised lines 45-49]

10. L52-54; Change to "Machine learning-based methods for the detection of anthropogenic influence (DAI) have been shown..." –as there are slightly difference approaches in Barnes et al. (2019, 2020) and Sippel et al. (2020).

Done.

[Revised line 53]

11. L56-57; Given that these are relatively new methods for the climate community, it would be helpful to further clarify how the forced signal is detected from the ANN setup.

Revised. Original and revised sentences are in blue and red below, respectively.

Original:

"An artificial neural network (ANN) is trained to predict a proxy of external forcing (e.g. the year of the data) based on the spatial maps of the target variable from an ensemble of GCM simulations. Then a forced signal can be confirmed despite the presence of internal climate variability and inter-model variability^{29,30}."

Revised:

"An artificial neural network (ANN) is trained to predict a proxy of external forcing (e.g. the year of the data) based on the spatial maps of the target variable from an ensemble of GCM simulations. Under this supervised learning approach, the ANN learns the spatial patterns that best represent the external forcing from the background noise arising from the internal variability and model uncertainty^{33,34}. Observations can then be fed to this trained ANN to assess the presence of an anthropogenic signal in observations³³⁻³⁵."

[Revised lines 55-60]

12. L89-91; Be clear here on the use of the word "model" which can be used to describe the ANN (e.g., in computer science) or GCM(e.g., in climate science)

Done. Revised as below (also taking the comment by reviewer #3 into account).

Original: "According to the models, the anthropogenic signal has probably already emerged in Rx1day, consistent with traditional statistical methods⁴⁰."

Revised: "The ANN suggests that there is a detectable anthropogenic signal in the GCM's Rx1day during the historical period, consistent with traditional statistical methods⁴⁴."

[Revised lines 93-94]

13. L92-94; I am not sure I understand why you picked the 1920-1949 period to composite for the LRP map here? Wouldn't it be more interesting to compare that base period to the LRP composite when the forced change is detected? I actually found the LRP result in Figure S4a more interesting than the one in the main manuscript.

Agreed. Changed the period of LRP maps to 1982-2015 in figure 1 [Revised line 159] and moved the *time varying fingerprints* section (including previous Figure S4a) to the main text (as the new Figure 2) [Revised lines 127-150].

14. L99-115; I think it would strengthen the manuscript to better contrast potential physical mechanisms that may relate to the LRPmaps. In fact, if there is room in the word count, I suggest moving some of the text from the supplement into this section.

Done. As mentioned in the previous answer, *time varying fingerprints* was moved to the main text.

[Revised lines 127-150]

15. L104-105; Where is this shown?

This sentence was removed.

16. L136; This section is an interesting result of these new methods that should be noted further in the conclusions/abstract!

This section was added to explain the reasons behind the spread and the outliers in the predicted year time series. However, it is difficult to frame this under the **detection** framework and therefore does not fit in the abstract/conclusion. It could, however, have other possible uses in *emergent constraint* research, to which we alluded in lines 186-187 (we added an extra reference there as well).

17. L145; What do you mean by “higher value?”

Changed “a higher value” to “a later year”.

[Revised line 183]

18. L162-163; State the years of the historical period.

Done. Added the period 1982-2015 as below. New additions are in red.

“In observations and reanalyses, only seven out of the eleven datasets show a significant trend ($p < 0.01$) in globally-averaged Rx1day for the historical period 1982-2015, ranging from 0.02 to 0.09 mm/day/year (Table S3).”

[Revised lines 203-206]

19. L164-166; This transition makes it sound like you are inputting maps of extreme precipitation trends into the ANN.

This paragraph was revised to explain why we are talking about the trend in observations. Newly added sections are in red below.

“With these physical interpretations of the ANN results and relevance patterns, we use the GCM-trained ANNs to detect whether there is a forced signal in observations. According to previous studies, a steady global warming trend can be seen since the 1970s⁵¹ and, in GCMs, the anthropogenic signal of global-mean Rx1day has started to emerge as of the 1970s⁴⁴. Therefore, according to the theoretical basis of the response of extreme precipitation to warming³⁻⁵, one could hypothesize that GCM-simulated and observed

Rx1day should have a positive significant trend during the historical period analyzed here, 1982-2015. Confirming this, GCMs show a positive trend in globally-averaged Rx1day (significant at 99% in 36 out of 44 models), which cannot be explained by natural variability alone (Figure S3). In observations and reanalyses, only seven out of the eleven datasets show a significant trend ($p < 0.01$) in globally-averaged Rx1day for the historical period 1982-2015, ranging from 0.02 to 0.09 mm/day/year (Table S3). Taken at face value, this large disparity in observations suggests that the observational evidence for anthropogenic influence on recent changes in extreme precipitation is weak. However, when we apply the ANN trained on Rx1day data from GCMs, to the same eleven datasets, a different story emerges.”

[Revised lines 196-209]

20. L171; “... of the ANN with different combinations of training and model GCM data.”
“random iterations of the ANN with different training/testing model sets”

Revised as below. Also added a sentence (shown below) in the Methods section to make it clearer.

Original: “from 51 random iterations of the ANN with different training/testing model sets.”

Revised: “from 51 different ANNs trained using randomly selected training GCMs.”

[Revised lines 214-215]

Newly added below sentence to the methods to make it clearer :

“We repeated the training process for 51 different training sets obtained by random combinations of GCMs, resulting in 51 different ANNs.”

[Revised lines 365-366]

21. L174; “by [random] chance.”

This sentence was removed because we replaced the randomly shuffled dataset (the null hypothesis) with pre-industrial control simulations. More details on this change can be found in the methods.

22. L175-176; What are in line? Their slopes?

Yes, their slopes. To make it clearer, added “the slopes” to the beginning of the sentence as below (newly added words are in red).

“When looking at the slope, two observational datasets (MSWEP and GPCC) are in line with GCMs”

[Revised lines 222-223]

23. 178; “reanalysis”

Corrected by changing “renalysis” to “reanalysis”.

[Revised line 225]

24. L185-186; But can we directly infer this from the negative relevance? Since LRP only tells us the areas of importance for the ANN to make a prediction, it could be the case that nonlinear combinations of different regions (positive + negative relevance) act to oppose/enhance trends.

We agree with this conceptual explanation. In this sentence, however, we are writing about the signal estimated as the trend in observations being suppressed by the noise (which corresponds to the negative relevance). To make it clearer, the sentence was revised as below.

Original: “In particular, areas of negative relevance, defined previously, can act to suppress the trend in the global mean.”

Revised: “In particular, areas with higher internal variability can act to suppress the trend in the global mean.”

[Revised lines 239-240]

25. L189-192; How does the skill of the ANN change if you remove the annual mean from each Rx1day map (e.g., as done in Barnes et al. 2019)? Thus, this could be a way to reduce the model mean state biases.

To test the method using Rx1day with **1) mean removed** and **2) locally standardized** (as suggested by reviewer#2) as input to the ANN, we redid the whole analysis, twice, for the two new inputs types. As shown in the figure below (figure R1), the median departure year for the above two cases (figures R1b and c) occur much later than the case when Rx1day is used without any modifications (figures R1a and the results in the manuscript). For instance, the 5th percentile of the distribution in figures R1b and c (denoted by the extent of the bottom whisker) is closer to the year 2000 while the mean (denoted by red horizontal line) is closer to or later than 2020. Therefore, according to models, we can expect a significant slope in the predicted year only after we are well into the 21st century. This makes the detection method presented in the manuscript using the trend and the correlation of the predicted year for the period unfeasible, as observational coverage is limited to 1982-2015. Removal of the mean change delaying the detection time and therefore not being applicable for detecting the anthropogenic influence in the historical record has been shown in the previous detection and attribution study Santer et al., (2007) (their table 1).

However, this exercise indicates that,

a) by increasing the temporal resolution (e.g. using seasonal, monthly or pentad based extremes instead of annual maximum)

and

b) using a different proxy of anthropogenic influence such as global mean surface temperature and a threshold-based detection method as in Sippel et al., (2020),

we could still use this machine learning based detection framework for mean removed or locally standardized extreme precipitation data.

Figure R1. Same as figure 1c but for three input types, **a)** area weighted Rx1day (the main analysis), **b)** global mean removed Rx1day, **c)** locally standardized Rx1day. Results are shown for all 51 random iterations and all models for each case.

26. L215; What do you mean by realistic radiative forcing?

It should be changed to “for historical simulations and future projections” as below (strikethrough was used for removed words and new additions are in red).

“Using a recently introduced ANN DAI method, ~~which utilizes~~ we utilized the time evolution of spatial maps of Rx1day in GCMs, ~~subject to realistic radiative forcing~~ for historical simulations and future projections.”

[Revised lines 269-270]

27. L229; Note that this is on a global mean scale.

We newly added the words in red below to indicate that our domain/scale is global.

“Therefore, the detected signal in multiple global terrestrial Rx1day datasets in this study....”

[Revised lines 284-285]

28. L397-398; Do you think that this relative coarse grid could affect the results of your ANN design for extreme precipitation?

We do not think that the coarse grid used here has a significant impact on our results because,

- i. We need a resolution small enough so that the number of features (predictors, in this case values at each grid cell) is not too large, compared to the number of samples (e.g. number of models and the length of the simulations) so that the regression problem is well posed. This is the so called “curse of dimensionality” issue in machine learning. At the same time, as the reviewer correctly pointed out, the resolution should be high enough to capture Rx1day well. For this reason, we selected the 2° resolution,

considering the resolution used in previous studies (referenced in the manuscript) with a GCM based analysis and a comparison with observations.

- a. Pfahl et al., (2017) : used $2^{\circ} \times 2^{\circ}$
- b. O’Gorman. (2012) : used $3^{\circ} \times 3^{\circ}$
- c. Previous D&A studies using HadEX datasets, e.g. Paik et al., (2020): used $5^{\circ} \times 5^{\circ}$

- II. Even in the current 2° resolution setup, we are altering the L2 regularization to account for the spatial autocorrelation/spatial structure of extreme precipitation systems. If one were to do the same exercise at a higher resolution (provided that enough data is being used as pointed out in the I. above), we would expect a larger L2 value which would provide similar spatial extents to a larger degree.

29. L418; Multicentury?

Here we meant longer simulations (multiple centuries) would suit better. But to avoid the confusion, we removed the word “multicentury” from the sentence (strikethrough was used for removed words below).

Revised: “By feeding the ANN ~~multicentury~~ data from forced simulations, it learns to distinguish the forced signal from internal climate variability.”

[Revised lines 346-347]

30. L423-424; Do you include polar regions, such as Antarctica?

No, we do not include Antarctica. This was added to the end of the sentence as below.

Revised: “Our primary goal is to detect the anthropogenic signal in extreme precipitation over land (excluding Antarctica).”

[Revised lines 351-352]

31. L432-436; Reference would be helpful here for readers unfamiliar with L2 regularization.

Added the following reference:

Hastie, T. (2020). Ridge regularization: An essential concept in data science. *Technometrics*, 62(4), 426-433.

[Revised line 360]

32. L347; Rephrase this sentence, as it reads contradictory to your statement in L439.

[The line number reviewer has provided here is a reference in the reference list. We believe they are referring to L437, instead of L347 and answered accordingly].

Done. We adopted a more robust training method with a validation dataset, and therefore modified this section as below. This was done in part to answer review questions regarding results with different types of inputs (e.g. mean removed, locally standardized).

Original: “We could have made the ANN more complex to achieve higher accuracy. But we elected not to do so, partly because Increasing the number of hidden units or changing the other hyperparameters (except for L2 regularization) did not result in a substantial increase

in accuracy. More importantly, we aimed to keep the ANN simple, with a reasonable degree of accuracy. This is because the main goal is not to obtain a perfect prediction, but rather to reveal the forced patterns the ANN learns (e.g. ref. 29, 30). As we show in Section 3, imperfections in the prediction also can be physically interpreted within the D&A research framework (Figure 2)."

Newly added: "We trained the model for 1000 epochs. *Early stopping* was enabled to reduce the overfitting by monitoring the *validation loss* with a *patience value* of 50 epochs⁸⁰. We repeated the training process for 51 different training sets obtained by random combinations of GCMs, resulting in 51 different ANNs. We found that increasing the number of hidden units or changing the other hyperparameters did not result in a substantial increase in accuracy."

[Revised lines 363-368]

Figures/Tables:

1. Figure 1; The color map needs to be changed in (a) and (b) to a colorblind friendly choice, instead of "jet/rainbow." For example, see Hawkins, 2015. The red dashed (1:1) line is also very difficult to see. There is a typo in the title under (a) for "training".
Done.
2. Figure 1; The caption for (d) seems different than the title. What LRP years are you compositing here?
Changed the time period used here to 1982-2015. Caption was corrected accordingly.
3. Figure 2; Aren't the relevance map unitless? I am a bit confused by the title under Figure 2a for (years).
LRP backpropagates the final output of the ANN, with the physical unit (in this case the year). If the output is a probability value (e.g. in a classification), then the LRP maps would be unitless.
4. Figure 3; I found it difficult to compare the observations in Figures 3a-d due to the different y-axis limits.
To resolve this issue, in the revised manuscript, we normalized each predicted year time series to have a common y axis. Results without normalizing are also included in supplementary (Figure S4).

References:

Bindoff, N. L., et al. (2013), Detection and attribution of climate change: From global to regional, in Climate Change 2013: The Physical Science Basis. Contribution of Working Group I to the Fifth Assessment Report of the Intergovernmental Panel on Climate Change, edited by T. F. Stocker et al., pp. 867– 952, Cambridge Univ. Press, Cambridge, U. K., and New York.

- Dong, S., Sun, Y., & Li, C. (2020). Detection of human influence on precipitation extremes in Asia. *Journal of Climate*, 33(12), 5293-5304.
- Dong, S., Sun, Y., Li, C., Zhang, X., Min, S. K., & Kim, Y. H. (2021). Attribution of extreme precipitation with updated observations and CMIP6 simulations. *Journal of Climate*, 34(3), 871-881.
- Hawkins, E. (2015). Scrap rainbow colour scales. *Nature*, 519(7543), 291-291.
- O'Gorman, P. A. (2012). Sensitivity of tropical precipitation extremes to climate change, *Nature Geoscience*, 5(10), 697– 700, doi:10.1038/NCEO1568.
- Paik, S., Min, S. K., Zhang, X., Donat, M. G., King, A. D., & Sun, Q. (2020). Determining the anthropogenic greenhouse gas contribution to the observed intensification of extreme precipitation. *Geophysical Research Letters*, 47(12), e2019GL086875.
- Paik, S., Min, S. K., Zhang, X., Donat, M. G., King, A. D., & Sun, Q. (2020). Determining the anthropogenic greenhouse gas contribution to the observed intensification of extreme precipitation. *Geophysical Research Letters*, 47(12), e2019GL086875.
- Pfahl, S., O'Gorman, P. A., & Fischer, E. M. (2017). Understanding the regional pattern of projected future changes in extreme precipitation. *Nature Climate Change*, 7(6), 423– 427. <https://doi.org/10.1038/nclimate3287>.
- Santer, B. D., Mears, C., Wentz, F. J., Taylor, K. E., Gleckler, P. J., Wigley, T. M. L., ... & Wehner, M. F. (2007). Identification of human-induced changes in atmospheric moisture content. *Proceedings of the National Academy of Sciences*, 104(39), 15248-15253.
- Sippel, S., Meinshausen, N., Fischer, E. M., Szekely, E., & Knutti, R. (2020). Climate change now detectable from any single day of weather at global scale. *Nature Climate Change*, 10, 35– 41. <https://doi.org/10.1038/s41558-019-0666-7>.
- Swain, D. L., Singh, D., Touma, D., & Diffenbaugh, N. S. (2020). Attributing extreme events to climate change: a new frontier in a warming world. *One Earth*, 2(6), 522-527.

Reviewer #2 (Remarks to the Author):

In this manuscript, Madakumbura et al. use a machine learning method to identify the anthropogenic influence on changes in extreme precipitation at the global scale. This is an important topic and this paper adds to the literature by using an entirely different method than most existing studies. I think it will be useful to those interested in either/both attribution and extreme precipitation. Given that what is new about this paper is in the details and the method, it could perhaps be better suited for a longer-form article. I would be interested to read more in-depth discussion of the results, including of the time-varying fingerprints. I have some comments below for the authors to consider.

We are extremely grateful to the reviewer for the valuable comments given, especially regarding how to improve the research framework/analysis presented under the theme of *detection (and attribution)*. We answered all the review comments and revised the manuscript accordingly. Please find the point by point answers below.

The introduction mentions two studies detecting human influence on increases in extreme precipitation. However, there are some other papers published in the last year that should be acknowledged:

- I. Paik et al. 2020: Determining the anthropogenic greenhouse gas contribution to the observed intensification of extreme precipitation. doi:10.1029/2019GL086875.
- II. Dong et al. 2021: Attribution of extreme precipitation with updated observations and CMIP6 simulations. doi:10.1175/JCLI-D-191017.1.
- III. Dong et al. 2020: Detection of human influence on precipitation extremes in Asia. doi:10.1175/JCLI-D-19-0371.1.

In addition to North America as already mentioned, anthropogenic influence has also been identified for Europe and Asia. I think the Paik et al. paper is cited in the supplementary material, but it should also be noted in the main text.

Done. Revised the sentence as below.

Revised sentence:

“Recent studies have detected anthropogenic influence in historical changes to extreme precipitation across the domains North America^{17,18}, Europe^{18,19}, Asia¹⁸⁻²⁰ and Northern Hemisphere land areas as a whole²¹.”

[Revised lines 35-37]

References in the above sentence (in the same order as the revised manuscript):

17. Kirchmeier-Young, M. C., and X. Zhang (2020), Human influence has intensified extreme precipitation in North America. *Proceedings of the National Academy of Sciences of the United States of America*, 117(24), 13,308–13,313, doi:10.1073/PNAS.1921628117.
18. Dong, S., Sun, Y., Li, C., Zhang, X., Min, S. K., & Kim, Y. H. (2021). Attribution of extreme precipitation with updated observations and CMIP6 simulations. *Journal of Climate*, 34(3), 871-881.
19. Paik, S., Min, S. K., Zhang, X., Donat, M. G., King, A. D., & Sun, Q. (2020). Determining the anthropogenic greenhouse gas contribution to the observed intensification of extreme precipitation. *Geophysical Research Letters*, 47(12), e2019GL086875.
20. Dong, S., Sun, Y., & Li, C. (2020). Detection of human influence on precipitation extremes in Asia. *Journal of Climate*, 33(12), 5293-5304.

21. Min, S. K., X. B. Zhang, F. W. Zwiers, and G. C. Hegerl (2011), Human contribution to more-intense precipitation extremes, *Nature*, 470, 378– 381, doi:10.1038/nature09763

How would the results be impacted if Rx1day was locally-standardized first? This could effect the differences between the wet and dry regions. Also, for the global means calculated, averaging non-standardized values will give more weight to the wet regions. Specific comments:

To test the method using Rx1day with **1) mean removed** (as suggested by reviewer#1) and **2) locally standardized** as input to the ANN, we redid the whole analysis, twice, for the two new inputs types. As shown in the figure below (figure R1), the median departure year for the above two cases (figures R1b and c) occur much later than the case when Rx1day is used without any modifications (figures R1a and the results in the manuscript). For instance, the 5th percentile of the distribution in figures R1b and c (denoted by the extent of the bottom whisker) is closer to the year 2000 while the mean (denoted by red horizontal line) is closer to or later than 2020. Therefore, according to models, we can expect a significant slope in the predicted year only after we are well into the 21st century. This makes the detection method presented in the manuscript using the trend and the correlation of the predicted year for the period unfeasible, as observational coverage is limited to 1982-2015. Removal of the mean change delaying the detection time and therefore not being applicable for detecting the anthropogenic influence in the historical record has been shown in the previous detection and attribution study Santer et al., (2007) (their table 1).

However, this exercise indicates that,

- I. **by increasing the temporal resolution (e.g. using seasonal, monthly or pentad-based extremes instead of annual maximum)** and
- II. **using a different proxy of anthropogenic influence instead of the year (e.g. global mean surface temperature) and a threshold-based detection metric as in Sippel et al (2020),**

we could still use this machine learning based detection framework for mean removed or locally standardized extreme precipitation data.

Figure R1. Same as figure 1c but for three input types, **a)** area weighted Rx1day (the main analysis), **b)** global mean removed Rx1day, **c)** locally standardized Rx1day. Results are shown for all 51 random iterations and all models for each case.

- Line 40: Can you please clarify or rephrase “trend of the projection”? Additionally, the optimal fingerprinting regression and the EOF-based fingerprinting approach as in ref 21 are different methods, with some different assumptions.

As the reviewer correctly pointed out, the original sentence only reflects the EOF based (e.g. Marvel and Bonfils, 2013) and machine learning based (e.g. Sippel et al., 2020) detection methods. For optimal fingerprinting, the fingerprint itself could be chosen as the trend of the GCMs and the resultant regression coefficients (i.e. projecting the observations on to the fingerprints) are evaluated for the signal detection.

We revised the sentence as below by removing the words “trend of the projections” and with additional references to the sentence.

Original: “Projection of observations onto these fingerprints allows for signal detection as the trend of the projection²¹.”

Revised: “Projection of observations onto these fingerprints allows for detection of the signal (Hegerl and Zwiers, 2011; Marvel and Bonfils, 2013)”

[Revised line 40]

- Line 81: Can you state either the analysis period or which period the first 7-8 decades are here?

Done. Revised the sentence as below (strikethrough was used for removed words and new additions are in red).

“Predictions of the simulated Rx1day year (Figure 1a,b) show that the ANN struggles during roughly the ~~first seven to eight decades of the analysis~~ 1920-1970 period.”

[Revised lines 83-84]

- Figure 1b: Why is the orange model so different from the others?

As described in the section *Origins of the spread in the predicted year*, that model and models with a low predicted year have a dry bias in regions that have a high positive relevance, such as African, Asian and South American monsoon regions. A similar result and an explanation can be found in Barnes et al., (2018) but for temperature (section 3.1, last paragraph). Note that in the revised manuscript, Fig 1b does not have that outlier as that model is in the training dataset (not in the testing dataset) in the shown training/validation/testing (random) model selection. As the detection analysis (Figure 4) is from 51 such random iterations of training/validation/testing models, results are not sensitive to the outliers.

- Figure 1d and the corresponding discussion could use clarifying. From both the subplot title and Line 92, I was left wondering how you could consider this the anthropogenic fingerprint when it is based on such an early period. The caption mentions 2070-2099, but this should be noted in the text too.

Done. This was corrected in Figure 1 (and in the text) by considering the historical observational period analyzed, 1982-2015.

- Line 97: It would helpful to say explicitly why using a time-based metric tells you about the anthropogenic forcing

A new sentence was added as below (new addition are in red).

“By learning how to predict the year of the data, the ANN is able to detect the spatial patterns that best reflect the changing climate from background noise^{33,34}. Therefore, the relevance patterns observed above can be considered as the ANN-identified fingerprints of anthropogenic influence on Rx1day (e.g. ref. 35).”

[Revised lines 100-103]

- Line 164: Why do these trends imply the anthropogenic forcing is weak? Also, what about the historical trends over this shorter period tells us the anthropogenic influence?

- I. Here we are saying that these disparate observational results suggest “**the evidence for anthropogenic influence on recent changes in extreme precipitation is weak**”, not that the anthropogenic influence/forcing itself is weak. To make it clear, we revised the sentence (shown in red below).

- II. Regarding the question “what about the historical trends over this shorter period tells us the anthropogenic influence”, we added an explanation (shown in green below) with an additional figure (Figure S3).

Fully revised paragraph based on this comment is shown below. New additions mentioned in (I) and (II) above are in red and green, respectively. Unchanged sections are in blue and strikethrough was used for removed words.

“With these physical interpretations of the ANN results and relevance patterns, we use the GCM-trained ANNs to detect whether there is a forced signal in observations. According to previous studies, a steady global warming trend can be seen since the 1970s⁵¹ and, in GCMs, the anthropogenic signal of global-mean Rx1day has started to emerge as of the 1970s⁴⁴, Therefore, according to the theoretical basis of the response of extreme precipitation to warming³⁻⁵, one could hypothesize that GCM-simulated and observed Rx1day should have a positive significant trend during the historical period analyzed here, 1982-2015. Confirming this, GCMs show a positive trend in globally-averaged Rx1day (significant at 99% in 36 out of 44 models), which cannot be explained by natural variability alone (Figure S3). ~~First, we calculate the globally-averaged Rx1day trends in each dataset using a modified Mann-Kendall trend test⁴³.~~ **In observations and reanalyses**, only seven out of the eleven datasets show a significant trend ($p < 0.01$) in globally-averaged Rx1day for the historical period 1982-2015, ranging from 0.02 to 0.09 mm/day/year (Table S3). Taken at face value, this **large disparity in observations** suggests that the **observational** evidence for anthropogenic influence on recent changes in extreme precipitation is weak. However, when we apply the ANN trained on Rx1day data from GCMs, to the same eleven datasets, a different story emerges.”

[Revised lines 196-209]

- Line 169: Can you please clarify why this needs to be just a positive slope and not a slope consistent with 1?

If the proxy of the anthropogenic forcing (the label) is something directly proportional to the anthropogenic forcing (e.g. global mean surface temperature or the radiative imbalance, as used in Sippel et al., 2020), the predicted vs actual time series should have a 1:1 line. In this case the detection could be defined as when the predicted proxy value continuously exceeds a threshold defined by the natural variability.

In our case, when the year is used as the proxy of the anthropogenic forcing, as it is not directly proportional to the anthropogenic forcing, we get a curve instead of a 1:1 line. This may or may not be parallel to (or overlap) the 1:1 line during the historical period.

However, we understand the issue of not having a well-defined measure here for the detection. As a more robust method of detection, we redid the “null hypothesis” component using pre-industrial control simulation for 220 non-overlapping 34-year segments (for the historical period 1982-2015) (Methods and Figure 4). This allows us to calculate a measure of noise from the natural variability. Using this noise measure, we can calculate signal-to-noise ratio and a measure of statistical significance (following the methodology in Marvel and Bonfils, 2013 and Marvel et al., 2015). More details can be found in methods [revised lines: 317-324] and results [revised lines: 229-235].

References:

- Barnes, E. A., Hurrell, J. W., Ebert-Uphoff, I., Anderson, C., & Anderson, D. (2019). Viewing forced climate patterns through an AI Lens. *Geophysical Research Letters*, 46, 13,389– 13,398. <https://doi.org/10.1029/2019GL084944>
- Hegerl, G., and F. Zwiers (2011), Use of models in detection and attribution of climate change, *Wiley Interdisciplinary Reviews: Climate Change*, 2(4), 570– 591.
- King, A. D., Donat, M. G., Fischer, E. M., Hawkins, E., Alexander, L. V., Karoly, D. J., et al. (2015). The timing of anthropogenic emergence in simulated climate extremes. *Environmental Research Letters*, 10(9), 94015. <https://doi.org/10.1088/1748-9326/10/9/094015>
- Marvel, K., & Bonfils, C. (2013). Identifying external influences on global precipitation. *Proceedings of the National Academy of Sciences*, 110(48), 19301-19306.
- Marvel, K., Zelinka, M., Klein, S. A., Bonfils, C., Caldwell, P., Doutriaux, C., ... & Taylor, K. E. (2015). External influences on modeled and observed cloud trends. *Journal of Climate*, 28(12), 4820-4840.
- O’Gorman, P. A., and T. Schneider (2009), The physical basis for increases in precipitation extremes in simulations of 21st century climate change, *Proceedings of the National Academy of Sciences of the United States of America*, 106(35), 14,773–14,777, [doi:10.1073/pnas.0907610106](https://doi.org/10.1073/pnas.0907610106).

Rahmstorf, S., Foster, G., & Cahill, N. (2017). Global temperature evolution: recent trends and some pitfalls. *Environmental Research Letters*, 12(5), 054001.

Sippel, S., Meinshausen, N., Fischer, E. M., Szekely, E., & Knutti, R. (2020). Climate change now detectable from any single day of weather at global scale. *Nature Climate Change*, 10, 35– 41. <https://doi.org/10.1038/s41558-019-0666-7>

Reviewer #3 (Remarks to the Author):

Review Madakumbura et al.

Key results

The authors use a relatively new machine learning technique, ANN, to find an anthropogenic signal in precipitation for multiple observation and reanalysis datasets, that accounts for internal variability and model uncertainty.

Validity

I am not aware of data/analysis related flaws that would prevent publication.

Significance

Results are significant.

Data and methodology

The method is applied to annual-maximum daily precipitation over land, and relies on changes in spatial patterns. My understanding is that ANN can better recognize the year of occurrence if there is anthropogenic forcing than if there is not because of the additional signal that can be learned.

I have limited background on both the methods and most of the numerous datasets used, as noted below.

Analytical approach

No comment. See below.

Suggested improvements See

below.

Clarity and context

There are several instances where I felt the clarity of the text could be improved to make this accessible to more readers. I have detailed these among my more specific comments provided below, but perhaps the most important is being more explicit about why ANN can identify the anthropogenic signal. I found that I required multiple reads, and comprehensive use of some of referenced articles to understand parts of this article.

References

Acceptable, to the best of my knowledge.

Your expertise

My primary expertise is in environments and dynamics of deep convection & extreme precipitation processes, specifically on meso- and convective scales. The specific details of the ANN method, GCMs, reanalysis, and global precipitation datasets are all outside of my expertise. Most of my commentary has focused on clarity/communication of findings.

We are extremely grateful to the reviewer for the very valuable comments provided, mainly regarding how to improve the clarity of the manuscript. We answered all the review comments and revised the manuscript accordingly. Please find the point by point answers below.

I have provided a number of more specific comments below, most of which focus on clarity and context:

1. L10,L12 “Global climate models produce large increases in extreme precipitation when subject to anthropogenic forcing..” and “Models produce diverse precipitation responses to anthropogenic forcing” seem somewhat contradictory.

Reworded as below to make it clear that although GCMs agree on the sign of change, the magnitude of future increases is highly variable across models (new additions are in red).

“The intensification of extreme precipitation under anthropogenic forcing is robustly projected by global climate models, but highly challenging to detect in the observational record. Large internal variability distorts this anthropogenic signal. Models produce diverse magnitudes of precipitation response to anthropogenic forcing, largely due to differing schemes for parameterizing subgrid-scale processes.”

[Revised line 13]

2. My takeaway was that changes in spatial patterns were really important to ANN’s ability to detect the changes, it would be nice if this could be worked into the abstract.

Done. Newly added parts to the abstract are in red.

“The intensification of extreme precipitation under anthropogenic forcing is robustly projected by global climate models, but highly challenging to detect in the observational record. Large internal variability distorts this anthropogenic signal. Models produce diverse magnitudes of precipitation response to anthropogenic forcing, largely due to differing schemes for parameterizing subgrid-scale processes. Meanwhile, multiple global observational datasets of daily precipitation exist, developed using varying techniques and inhomogeneously sampled data in space and time. Previous attempts to detect human influence on extreme precipitation have not incorporated model uncertainty, and have been limited to specific regions and observational datasets. Using machine learning methods that can account for these uncertainties and capable of identifying the time evolution of the spatial patterns, we find a physically interpretable anthropogenic signal that is detectable in all global observational datasets. Machine learning efficiently generates multiple lines of evidence supporting detection of an anthropogenic signal in global extreme precipitation.”

[Revised lines 19-20]

3. L56-59: Seems like three different versions of the same sentence (esp. the second and third)? Could they be combined?

Agreed. Revised as below.

Original two sentences pointed out by the reviewer as similar:

- I. “Then a forced signal can be confirmed despite the presence of internal climate variability and inter-model variability^{29,30}. ”
- II. “This method also has the advantage of being able to explicitly include internal variability and model uncertainty.”

Sentence (I). above was revised as below (incorporating a suggestion from reviewer #1).

“Under this supervised learning approach, the ANN learns the spatial patterns that best represent the external forcing from the background noise arising from the internal variability and model uncertainty^{33,34}. Observations can then be fed to this trained ANN to assess the presence of an anthropogenic signal in observations³³⁻³⁵. ”

[Revised lines 57-60]

4. L68: I believe the definition of Rx1day is annual-maximum daily precipitation?

Correct. Changed “annual daily maximum precipitation” to “annual maximum daily precipitation”.

[Revised line 71]

5. Would it be possible to use known dataset traits (e.g. data availability/regional biases/etc) along with your results to gain additional insights?

Unfortunately, not all these datasets provide dataset traits such as data quality, station density and other information they used to generate the dataset. However, our attempt in *Supplementary section 3. Source of the spread in the signal of observation* and supplementary figures 8 and 9. will be helpful in future studies that will conduct intercomparisons of data products (e.g. Bador et al., 2020, Alexander et al., 2020).

6. L89: “According to the models...” seems vague, I assume you mean in this application of your analysis method to GCM projections. Is what you are trying to say that because there is a detectable signal in Rx1day in GCMs for present day using this method, it should therefore be able to detect a Rx1day signal of change in obs/reanalysis? Perhaps consider editing for clarity...

Agreed. Revised as below.

Original: “According to the models, the anthropogenic signal has probably already emerged in Rx1day, consistent with traditional statistical methods⁴⁰.”

Revised: “The ANN suggests that there is a detectable anthropogenic signal in the GCM’s Rx1day during the historical period, consistent with traditional statistical methods⁴⁴”

[Revised lines 93-94]

7. L92: While it’s defined in the Data/Methods section, I’d suggest defining relevance pattern here at the first use. This is a new term to me (and I suspect to most readers).

Revised as below.

Original: “Figure 1d shows the relevance pattern identified by the ANN, averaged over the period ...”

Revised: “Figure 1d shows the importance of each grid box for the ANN to identify the anthropogenic signal (hereafter called *relevance patterns*, see methods), averaged over the period ...”

[Revised lines 95-96]

8. L94-96: Is there a simpler way to say ‘advancing tendency on the prediction (i.e. the year)’ ...? Replaced “advancing tendency” with “positive contribution” and “retreating tendency” with “negative contribution”. A recent work in under review (which is currently only available as a preprint) supports this interpretation (Mamalakis et al 2021, their figure 3).

Original: “Therefore, areas of positive relevance can be interpreted as the regions with an advancing tendency on the prediction (i.e. the year) and negative values are the regions with a retreating tendency.”

Revised: “Therefore, areas of positive relevance can be interpreted as the regions with a positive contribution to the prediction (i.e. the year) and negative values are the regions with a negative contribution.”

[Revised lines 98-99]

9. L103-108: While connection to physical processes is important, it’s placement here seems abrupt and I find the transitions into and out of it difficult to follow. Though maybe it’s just the ‘also’ in line 103 that’s throwing me off at the start.

I felt like I needed to go read parts of the referenced paper to understand what the authors were trying to say here, so perhaps a bit more detail/explanation can be provided to alleviate that need. Specifically, this is the first mention of ‘dynamic’ and ‘thermodynamic’ components, which have very specific definitions in the referenced paper...

Revised as below. The word “also” was removed. We added the definitions of the dynamic and thermodynamic components in the sentence as suggested.

Original: “Regions with negative relevance **also** coincide with areas exhibiting a large negative dynamical component of the Rx1day trend (ref. 41, their Figure 3b). These regions show a significant anthropogenic reduction of vertical velocities associated with Rx1day. This offsets the Rx1day increase stemming from the thermodynamic contribution, and produces only a weak and inconsistent increase in Rx1day⁴¹.”

Revised: “Regions with negative relevance coincide with areas where the dynamical component of the Rx1day trend (i.e. the contribution from the change in vertical velocity⁴) is largely negative (ref. 45, their Figure 3b). This offsets the Rx1day increase stemming from the thermodynamic contribution (i.e. the contribution from the increase in atmospheric moisture with warming³⁻⁵) and produces only a weak and inconsistent increase in Rx1day⁴⁵.”

[Revised lines 108-112]

10. L108: “As suspected,” why was this suspected? I think the transition between the previous discussion and the signal-to-noise ratio could be made a bit more explicit.

Agreed. Revised as below by adding a new sentence to the beginning for the transition between previous discussion and signal-to-noise ratio analysis. Also removed “as suspected” (original, unchanged sentences are in blue).

“To understand the physical nature of the relevance patterns, we next assess the signal, and the noise components arising from internal variability and the model uncertainty. ~~As suspected,~~ Negative relevance of the forced response is associated with a lower signal-to-noise ratio (S:N) than the regions with positive relevance (Figure 1e,f). The S:N is lower for both internal variability and model variability. This reflects both the higher uncertainty regarding the change in extreme precipitation projected by GCMs for a majority of global arid land regions, as well as larger internal variability in those regions.”

[Revised lines 114-119]

11. L119-120: “The selection of regions in these previous studies (e.g. ref 16,17) seems to overlap’ – could you provide specific examples of apparent overlap so that readers don’t need to compare figures across multiple papers?

Revised as below (original/unchanged parts are in blue and new additions are in red).

“The ANN-based relevance patterns are consistent with the idea that previously observed long-term trends of terrestrial Rx1day are anthropogenic in origin (e.g. ref. 21, their Figure 1e). Many wet land regions, such as the Asian, African and South American monsoon regions, have experienced a robust increase in Rx1day to date^{15,16}, whereas in arid and semi-arid subtropical zones no such trend can be seen¹⁶. The selection of regions in these previous studies (e.g. ref. 16) seems to overlap with regions of higher relevance in Fig.1d”

[Revised lines 120-125]

12. L145-146: Higher value of what?

Changed “a higher value” to “a later year”.

[Revised line 183]

13. What is ‘these physical interpretations’ referring to?

[Line number is missing in the review comment, but this is referred to original lines 159-160].

In this sentence, “these physical interpretations” refers to the physically meaningful nature of the,

- I. ANN’s relevance patterns (section: ANN-identified fingerprints of anthropogenic influence)
- II. reasons behind over/under prediction of the year (section: *Origins of the spread in the predicted year*).

In the manuscript, the subsections prior to this section (revised manuscript line 195) are aiming to explain the physical nature of above results.

14. Did you notice any specific themes in peaks (Fig. 3a-d,) where ANN way overpredicts or under predicts the year (e.g. volcanic events like described in the Barnes et al study)?

Yes, we can see a dip in the predicted year time series of observations (Figure 4a-d) coinciding with the eruption of Mount Pinatubo, in 1991. As pointed out in Barnes et al., (2019), this could be indicating the small number of such events in the training dataset and the difficulty of shallow ANN to learn such events. Furthermore, similar errors in identifying the predicted year could occur by the underestimation of the precipitation response to volcanic activities as well as natural variability such as ENSO in GCMs, as pointed out in previous detection and attribution (D&A) studies (Zhang et al., 2007). This can be seen in a previous D&A attempts of precipitation as well (e.g. figure 3a of Marvel and Bonfils, 2013).

We included these caveats in a new paragraph (shown below) and added to the section conclusion.

“Several caveats of the machine learning based detection method should be noted. Compared to regression based traditional D&A methods (e.g. ref. 59), the assessment of the influence of individual forcings (e.g. anthropogenic aerosols, land-use change, and natural forcings such as volcanic and solar activities) in the presented framework is challenging. We did not attempt such a breakdown in this study, and this would require methodological modifications⁶⁰. Additionally, the training GCMs might be under sampling the low frequency natural variability such as Atlantic Multidecadal variability and Pacific Decadal Oscillation. This may be remedied by inflating the training dataset with paleoclimate data⁶¹. However, underestimation of the precipitation response to natural forcings such as volcanic activities and natural variability such as El Nino Southern Oscillation in GCMs could affect the results⁶².”

[Revised lines 291-299]

15. Am I correct in understanding that it’s actually the change in the ability of ANN to successfully predict the year over time that supports the finding of anthropogenic influence? In other words, this shallow version of ANN can not distinguish years based on spatial patterns of annual-maximum daily rainfall alone, until the impacts of anthropogenic forcing are seen, and the time that this occurs is considered the departure year...? It might be good to explicitly say this, maybe between the two sentences in line 82, since this is still quite a new technique (and it took me multiple reads & accessing Barnes et al. to sort this out). Is there a situation where it would be possible for ANN to successfully predict the years based on spatial patterns of Rx1day in the absence of anthropogenic forcing?

If the ANN (or any other type of ML model) does not overfit (i.e. generalize well), we should expect this characteristic, a near constant predicted year until the signal of climate change emerges from the noise of natural variability (Hawkins and Sutton, 2012). This characteristic of the predicted year suggests that the ANN is learning to identify the patterns of the forced response when it emerges.

Regarding the question whether the ANN can successfully predict the year in the absence of anthropogenic forcing, the answer is no (given that the ANN does not overfit). As pointed out above, this is because there are no spatial patterns for the ANN to learn which can represent the proxy of the anthropogenic forcing, the year, which is a monotonically increasing

variable. If the proxy is something non monotonic, such as global mean surface temperature (GMST), then it is possible to train an ANN that could predict the GMST even in absence of anthropogenic forcing. In this case the ANN learns the sensitivity of extreme precipitation to temperature.

This answer is supported by a new analysis conducted during the revisions. To make the analysis more robust, instead of shuffled models in Figure 4, we redid the analysis using 34-year non overlapping pre-industrial control (piControl) simulations (see methods). Results are shown in Figure 4, where the slope and r values for piControl are near zero. This is due to the fact in absence of external forcing the ANN cannot correctly predict the year.

As suggested, we newly added below sentence (in red).

“Predictions of the simulated Rx1day year (Figure 1a,b) show that the ANN struggles during roughly the 1920-1970 period. But prediction accuracy gradually increases, noticeably starting from the late 20th century. This characteristic, a near constant predicted year followed by a positive trend, is consistent with the emergence of the anthropogenic signal from the noise of natural variability⁴³.”

[Revised lines 83-87]

16. Fig 3 caption: What do you mean by randomly shuffling? Perhaps a better question, what is being shuffled?

In the revised manuscript, this analysis was replaced with results from a non-overlapping pre-industrial control simulation equal to the length of the observations. Details are added to the methods. This was done to provide a more robust measure of the detection and a measure of statistical significance.

17. Supplementary material 2nd to last paragraph: Could any known qualities about observational datasets explain something about these patterns (e.g. sparsity of obs)?

As wrote in the answer to comment#5, not all these datasets provide traits such as data quality, station density with the precipitation data. This makes the above task very difficult (may need a massive collaborative effort with dataset developers). However, a natural follow-up study would be to pinpoint how the observational uncertainty contributes to the difference in the signal and noise, as indicated by the reviewer’s comment here and in the perspective by Hegerl et al., (2015).

It might also be good to clarify here that you are no longer talking about the HadEX3 dataset, but the original observational datasets used...

Done. Added “For observational datasets used in the main text,” to the beginning of the paragraph.

References:

- Alexander, L. V., Bador, M., Roca, R., Contractor, S., Donat, M., & Nguyen, P. (2020). Intercomparison of annual precipitation indices and extremes over global land areas from in situ, space-based and reanalysis products. *Environmental Research Letters*, 15(5), 055002. <https://doi.org/10.1088/1748-9326/ab79e2>.
- Bador, M., Alexander, L. V., Contractor, S., & Roca, R. (2020). Diverse estimates of annual maxima daily precipitation in 22 state-of-the-art quasi-global land observation datasets. *Environmental Research Letters*, 15(3), 35005. <http://doi.org/10.1088/1748-9326/ab6a22>.
- Baines, P. G., & Folland, C. K. (2007). Evidence for a rapid global climate shift across the late 1960s. *Journal of Climate*, 20(12), 2721-2744.
- Barnes, E. A., Hurrell, J. W., Ebert-Uphoff, I., Anderson, C., & Anderson, D. (2019). Viewing forced climate patterns through an AI Lens. *Geophysical Research Letters*, 46, 13,389– 13,398. <https://doi.org/10.1029/2019GL084944>.
- Hawkins, E., & Sutton, R. (2012). Time of emergence of climate signals. *Geophysical Research Letters*, 39(1).
- Hegerl, G. C., Black, E., Allan, R. P., Ingram, W. J., Polson, D., Trenberth, K. E., ... & Zhang, X. (2015). Challenges in quantifying changes in the global water cycle. *Bulletin of the American Meteorological Society*, 96(7), 1097-1115.
- Mamalakis, A., Ebert-Uphoff, I., & Barnes, E. A. (2021). Neural Network Attribution Methods for Problems in Geoscience: A Novel Synthetic Benchmark Dataset. arXiv preprint arXiv:2103.10005.
- Marvel, K., & Bonfils, C. (2013). Identifying external influences on global precipitation. *Proceedings of the National Academy of Sciences*, 110(48), 19301-19306.
- Marvel, K., & Bonfils, C. (2013). Identifying external influences on global precipitation. *Proceedings of the National Academy of Sciences*, 110(48), 19301-19306.
- Sippel, S., Meinshausen, N., Fischer, E. M., Szekely, E., & Knutti, R. (2020). Climate change now detectable from any single day of weather at global scale. *Nature Climate Change*, 10, 35– 41. <https://doi.org/10.1038/s41558-019-0666-7>
- Zhang, X., Zwiers, F. W., Hegerl, G. C., Lambert, F. H., Gillett, N. P., Solomon, S., ... & Nozawa, T. (2007). Detection of human influence on twentieth-century precipitation trends. *Nature*, 448(7152), 461-465.

Reviewer comments, second round –

Reviewer #1 (Remarks to the Author):

Additional comments (R1):

In summary, this is a substantially improved manuscript. I thank the authors for their careful attention in addressing all of the Reviewers' comments and concerns.

Recommendation:

Accept to Nature Communications

Reviewer #2 (Remarks to the Author):

I am generally satisfied with the responses to my comments. I find the revised version to be improved; in particular, several places I found confusing in the initial manuscript are now more clear.

Reviewer #3 (Remarks to the Author):

Second Review of Madakumbra et al.

I appreciated reading the authors responses to both my and the other reviewers' comments. Thanks to the authors for their thoroughness, and for taking the time to explain things like my comment #15. I think the new draft represents a great improvement in readability of this interesting study.

At this point, I am satisfied that my comments have been addressed and can recommend accept providing the other reviewers are also satisfied that their concerns have been addressed, and after a few minor comments have been addressed.

Minor comments:

297-298: Is "However" the intended transition here? I was a little confused about the link between the two sentences.

Supplemental materials:

1. Perhaps check for consistency when describing ensembles and members. "We used 40 initial condition perturbed ensembles from the dataset for the period 1920-2099. Simulations follow similar forcing as CMIP5 models described in methods. To follow the same ANN training process as the first step, we used 26 members for training, 9 ensembles for validation and the rest (5) for testing. Thereafter, the analysis is identical to the main analysis (Figure S7)"

Reviewer #3 (Remarks to the Author):

Second Review of Madakumbra et al.

I appreciated reading the authors responses to both my and the other reviewers' comments. Thanks to the authors for their thoroughness, and for taking the time to explain things like my comment #15. I think the new draft represents a great improvement in readability of this interesting study.

At this point, I am satisfied that my comments have been addressed and can recommend accept providing the other reviewers are also satisfied that their concerns have been addressed, and after a few minor comments have been addressed.

We would like to thank the reviewer once again for taking time to review our manuscript. We answered the two new minor comments below.

Minor comments:

297-298: Is "However" the intended transition here? I was a little confused about the link between the two sentences.

Agreed. In the previous sentence, we mention a possible solution to the undersampling of the natural variability. But the model deficiencies, such as underestimation of the precipitation response to natural forcings and natural variability, could still affect the results. We revised the sentence by adding additional words as below. Original sentences are in blue and new additions are in red.

"Additionally, the training GCMs might be under sampling the low frequency natural variability such as Atlantic Multidecadal variability and Pacific Decadal Oscillation. This may be remedied by inflating the training dataset with paleoclimate data⁶¹. However, even with adequate sampling of natural variability in the training dataset, the underestimation of the precipitation response to natural forcings such as volcanic activities and natural variability such as El Nino Southern Oscillation in GCMs could still affect the results⁶²."

[Revised lines 297-302]

Supplemental materials:

1. Perhaps check for consistency when describing ensembles and members. "We used 40 initial condition perturbed ensembles from the dataset for the period 1920-2099. Simulations follow similar forcing as CMIP5 models described in methods. To follow the same ANN training process as the first step, we used 26 members for training, 9 ensembles for validation and the rest (5) for testing. Thereafter, the analysis is identical to the main analysis (Figure S7)"

Done. Changed "ensembles" to "ensemble members" as below.

"We used 40 initial condition perturbed ensembles members from the dataset for the period 1920-2099. Simulations follow similar forcing as CMIP5 models described in methods. To follow the same ANN training process as the first step, we used 26 ensemble members for training, 9 ensembles members for validation and the rest (5) for testing."